# Assessing the dimensionality of scores derived from the Revised Formal Thought Disorder Self-Report Scale in schizotypy

**Philip J. Sumner**[1]*, **Denny Meyer**[1], **Sean P. Carruthers**[1], **Fakir M. Amirul Islam**[2], **Susan L. Rossell**[1,3]

**1** Centre for Mental Health, Swinburne University of Technology, Melbourne, VIC, Australia, **2** Department of Health Science and Biostatistics, Swinburne University of Technology, Melbourne, VIC, Australia, **3** Department of Mental Health, St Vincent's Hospital, Melbourne, VIC, Australia

* psumner@swin.edu.au

## Abstract

The current work explored the dimensionality and convergent validity of responses to Barrera et al.'s (2015) 29-item Formal Thought Disorder–Self Scale (FTD-SS) obtained in two non-clinical samples. Exploratory factor analyses were conducted in Sample 1 (*n* = 324), yielding evidence of three correlated factors, although simple structure was not achieved until nine items were removed. Support for the correlated three factors model of responses to the revised 20-item scale (FTD-SS-R) was replicated when a confirmatory factor analysis was conducted in Sample 2 (*n* = 610). Finally, convergent associations were found between FTD-SS-R scores and scores from other schizotypy measures across both samples, though these measures only explained half of the variance in FTD-SS-R scores. Additional research is needed to evaluate the appropriateness of the items and incremental validity of the scale in non-clinical samples.

## Introduction

Schizotypy is a complex concept that links the variation in non-clinical behaviours and psychological experiences with the development of schizophrenia and schizophrenia-related disorders. As such, the concept incorporates dimensional and quasi-dimensional models of psychopathology with trait theories of personality [1–6]. The behaviours and experiences of schizotypy resemble the signs and symptoms of schizophrenia, only they tend to be less distressing than clinical symptoms and interfere less with the person's ability to function in daily life (e.g. [7–9]). Hence, these phenomena can occur in people who are not considered to have any psychiatric illness [5]. The relatively consistent occurrence of schizotypal behaviours and experiences in a person over time is described as a schizotypal personality [1], a typology that is often thought to reflect the maladaptive expression of one or more personality traits from the general population [2, 10–15]. Yet, schizotypal phenomena are also thought to share at least some of the same aetiologies as the signs and symptoms of schizophrenia [3–5, 16], and have been mapped according to the clinical syndromes of schizophrenia [1, 6, 13, 14, 17–20]. Thus, schizotypy is often considered to convey a predisposition for psychosis and is sometimes represented as a precursory or intermediary stage in its pathogenesis (e.g. [3, 5]).

**Data Availability Statement:** The participant informed consent information stated that demographic data will only be published in pooled format, to help protect anonymity. All other

relevant data have been attached as supporting information.

**Funding:** SLR was supported by an Australian National Health and Medical Research Council (NHMRC) Senior Research Fellowship (GNT1154651). URL: https://www.nhmrc.gov.au/. The funders had no role in study design, data collection and analysis, decision to publish, or preparation of the manuscript.

**Competing interests:** The authors have declared that no competing interests exist.

One of the more under-researched aspects of schizotypy is the presence of disorganised or constrained thought and speech. When clinically relevant, these phenomena are usually referred to collectively as 'thought disorder', 'formal thought disorder' or 'speech disorder', amongst various other names [21–23]. Such terms refer to the improper sequencing and expression of thought, which is evidenced through speech that seems odd or is difficult to understand. Moreover, speech that is unexpected, inappropriate or bizarre without diminishment in productivity or fluency (i.e. positive thought disorder) is often distinguished from underproductive or dysfluent speech (i.e. negative thought disorder). The severity of thought disorder reflects both frequency of occurrence and the extent to which communication is impaired, with severe thought disorder typically being associated with acute manic and psychotic states [24, 25]. In the context of schizotypy, milder manifestations of these phenomena are occasionally referred to as 'cognitive slippage' [26–30], though this term originally incorporated delusions and hallucinations in addition to disorganised and constrained thought and speech ([3]; see [31], p. 17). The term 'disorganized or constrained thought and speech' (DCTS) is used here to refer to these phenomena over the entire continuum, without connoting the presence of illness or impairment in the general population.

Despite relatively scant scientific interest, DCTS might be especially amenable to schizotypy research because it displays some trait characteristics. Firstly, prevalence and severity data strongly support the existence of a DCTS continuum that spans numerous psychiatric diagnoses and includes people without mental health disorders (e.g. [32, 33], see [24]). This dimensionality is incorporated into the operationalised definitions of severity used in many clinician-based rating scales, where the mildest detectable manifestations are defined either as non-pathological or as a questionable indication of impairment (e.g. [34–39], see [40]). Secondly, DCTS demonstrates co-familiality [32, 33]. That is, instances of DCTS tend to be more common amongst the unaffected first-degree relatives of people with schizophrenia compared to people without a family history of schizophrenia (e.g. [34, 35]). Finally, although severe state-based DCTS is often seen in acute psychotic or manic episodes [24, 25], residual signs can persist in some people despite treatment (e.g. [36–38]).

Researchers aiming to address this gap in the literature might consider using measures designed specifically for the assessment of DCTS. Most general schizotypy measures tend to devote only a few items to the assessment of disorganized speech, producing either single summary scores (e.g. [39]) or disorganization scores that conflate aspects of positive thought disorder with eccentric non-verbal behaviour, inattentiveness, social anxiety and other schizotypal phenomena (e.g. [40]). This is potentially problematic because DCTS is itself considered to be multidimensional, at least in clinical samples [41–52]. Accordingly, clinical measurement systems that have been designed purposefully to capture the various distinct manifestations of thought disorder tend to be more sensitive than global measures [53]. Yet, despite increased measurement sensitivity being particularly important for capturing the subtle variations in DCTS that are likely to characterize schizotypy, these measurement systems are mostly observer-rated, which can be time-consuming and require specialized training to administer [38], precluding their use in many research contexts.

Another potential way of assessing DCTS is to use self-report questionnaires. Questionnaires have the advantage of being quick and easy to administer, and so can be disseminated across relatively large samples. Questionnaires also rely upon a first-person perspective to capture introspected DCTS, producing measurements that presumably reflect some combination of phenomenological awareness and objective severity [54]. This represents an important justification for subjective measures of DCTS, including subjective interview-based measures (e.g. [46]) and self-report questionnaires, since they can serve to impose additional constraints

upon the testing of some hypotheses when used in combination with more objective measures, and thus provide a complementary avenue for research [46, 54, 55].

To date, three self-report questionnaires have been developed that are dedicated to the assessment of DCTS: the Cognitive Slippage Scale (CSS, [26]), the Communication Awareness Scale (CAS, [54]), and the Formal Thought Disorder–Self-Report Scale (FTD-SS, [56]). However, there is only preliminary psychometric evidence to support their use. Responses on all three questionnaires have demonstrated reliability in non-clinical samples (CSS: [13, 28, 29, 61], CAS: [58], FTD-SS: [60, 62]). Moreover, the scale scores produced from these responses appear to conform to the continuum model of psychopathology. For instance, self-report DCTS scores tend to be elevated amongst people with schizophrenia-related diagnoses compared to non-clinical samples (CSS: [57], FTD-SS: [58]), and amongst the first-degree relatives of people with schizophrenia compared to the first-degree relatives of people with non-psychotic psychiatric disorders (CSS: [30]). CSS scores obtained by children may also predict the level of schizotypy that they express in adulthood, as well as their likelihood of developing schizophrenia [27]. Finally, convergence has been demonstrated between scores from these questionnaires and responses on other schizotypy measures (CSS: [13, 28, 63], FTD-SS: [62]), with stronger associations found for more closely related aspects of schizotypy [13, 58], as well as between self-report DCTS scores and measures of executive cognitive functions (CSS: [13, 27], CAS: [58], FTD-SS: [62]).

Although promising, many of these psychometric findings need to be replicated and extended upon. In particular, it remains to be seen whether or not the multidimensionality of DCTS is adequately represented in the responses to these questionnaires. Several studies have included CSS total scores within a battery of schizotypy measures and used dimension-reduction analyses to explore any underlying latent variables [28, 29, 59]. However, by limiting the analyses to total scores, these studies overlooked the potential multidimensionality contained within the construct of DCTS itself. Only one study has explored the dimensionality of item-responses from a non-clinical sample using a self-report measure of DCTS; this study conducted a principal components analysis on FTD-SS responses and revealed evidence of three inter-related dimensions ([56], see Fig 2) indicative of a 'correlated traits model' [60]. The three dimensions, which were labelled 'odd speech', 'conversational ability' and 'working memory deficit' [56], upheld the common distinction between positive and negative thought disorder, and seemed to align conceptually with some factor solutions derived from clinician-rated scales in clinical samples (e.g. [50]).

If the FTD-SS is able to capture some of the multidimensionality of DCTS, then it could potentially be a more sensitive measure of DCTS than the general schizotypy questionnaires that are more commonly used. Therefore, the aim of the current work was to further investigate the construct validity of scores derived from the FTD-SS in non-clinical samples. In particular, exploratory factor analyses were applied to a sample of FTD-SS responses collected from university students to evaluate the underlying unidimensionality or multidimensionality. The replicability of the results of these exploratory analyses were subsequently tested in a second non-clinical sample using confirmatory factor analysis techniques. Finally, the convergence of FTD-SS scores with scores derived from other schizotypy measures was assessed, as well as the influence of demographic variables on FTD-SS scores.

## Method

### Participants

Two samples were collected in the current study. Participants in Sample 1 were students recruited through the Research Experience Programme (REP) at Swinburne University of

Technology, and were awarded course credit for their participation. Participants in Sample 2 also included students that were recruited via REP. However, to increase the representativeness of the second sample, participants were additionally recruited through Prolific [61], a crowdsourcing platform designed to connect participants with research studies, and friends, family and close colleagues were invited to participate using a snowballing recruitment strategy. Participants recruited through Prolific were financially reimbursed the AUD equivalent of £5 (GBP). All participants were screened based on their self-reported responses to three inclusion criteria. Participants had to be: 1) at least 18 years old; 2) current residents of Australia; and 3) free from any current diagnosed mental health disorders.

## Materials

Barrera et al.'s [56] Formal Thought Disorder–Self-Report Scale (FTD-SS) is a 29-item questionnaire that was designed to assess various difficulties with communication, including pragmatics, lexical selection and syntax, memory and attention during conversation, paralinguistic and non-verbal communication, and other signs of thought disorder [42]. Each item describes a specific communicative difficulty, with responses being made according to a four-point ordinal scale in terms of frequency of occurrence (ranging from 1 - "almost never" to 4 - "almost always"). Scores represent the sum of item responses, with higher scores indicating more frequent communication difficulties. Barrera et al. [56] reported coefficient alphas of 0.93 and 0.86 for total summed scores in two non-clinical samples. They also reported evidence of three inter-related dimensions, two consisting of seven items each (conversational ability: $\alpha = 0.87$; working memory deficit: $\alpha = 0.82$) and one consisting of 15 items (odd speech: $\alpha = 0.88$).

The Schizotypal Personality Questionnaire (SPQ) is a measure of schizotypy that was modelled on the DSM-III diagnostic criteria for schizotypyal personality disorder [39]. It contains 74 items that contribute to an overall total score and nine subscale scores, with the nine-item Odd Speech subscale specifically capturing disorganized (but not impoverished) thought and speech. Items depict schizotypal experiences, which are endorsed using a dichotomous response format (i.e. "yes" or "no"), with higher scores indicating greater levels of schizotypy. Evidence of internal consistency was found in the current sample (total scores: $\alpha = 0.97$; subscale scores: $0.87 \leq \alpha \leq 0.92$).

The Oxford-Liverpool Inventory of Feelings and Experiences (O-LIFE, [62]) was designed to comprehensively assess schizotypy, representing the culmination of numerous preceding personality- and symptom-based measures. It contains 104 items that contribute to four scales, with the 24-item Cognitive Disorganization scale encompassing elements of disorganized (but not impoverished) thought and speech, in addition to poor attention, concentration and decision-making, and social anxiety [40]. Items depict schizotypal experiences and elicit dichotomous responses (i.e. "yes" or "no") based on their endorsement by the respondent, with higher scores indicating greater levels of schizotypy. In the current sample, internal consistency was demonstrated for all four scales (cognitive disorganization scores: $\alpha = 0.94$; unusual experiences scores: $\alpha = 0.95$; introvertive anhedonia scores: $\alpha = 0.90$; impulsive non-conformity: $\alpha = 0.79$).

Finally, forty-eight items from the International Personality Item-Pool (IPIP, [63]) were used to assess neuroticism and extraversion. These items comprise part of a larger five-factor measure of personality [64]. The two personality domains each span six facets, with each facet consisting of four items. However, only the summed domain scores for neuroticism and extraversion were calculated for this study. The items represent personal descriptors and are endorsed according to a five-point Likert-type scale (1 = "very inaccurate", 2 = "moderately inaccurate", 3 = neither accurate nor inaccurate", 4 = "moderately accurate", and 5 = "very

accurate"), with higher scores denoting a greater level of trait expression. Internal reliability was demonstrated in the current sample (neuroticism total scores: $\alpha = 0.91$; extraversion total scores: $\alpha = 0.90$).

## Procedure

The measures were presented as part of a larger online survey. The FTD-SS was always presented after demographic questions, but the remaining schizotypy measures were presented after the FTD-SS in a randomized order. For Sample 1, the items of the FTD-SS were administered in the same prescribed order as reported by Barrera et al. [56]. For Sample 2, the order in which the FTD-SS items were presented was randomized. Participation in the survey was entirely anonymous, with implied consent being obtained. The study protocol was approved by the Swinburne Human Research Ethics Committee (2019/154).

## Analysis

Three groups of analyses were conducted: exploratory dimension-reduction analyses of responses to the FTD-SS obtained in Sample 1, a confirmatory factor analysis of responses to the FTD-SS obtained in Sample 2, and regression analyses between FTD-SS scores and scores derived from the other measures of schizotypy across both samples.

The exploratory dimension-reduction analyses conducted in Sample 1 involved an initial exploratory factor analysis, followed by an exploratory bifactor analysis and Rasch analysis. The results of these analyses then guided the exclusion of FTD-SS items from subsequent exploratory factor analyses until an adequate solution was found. The initial exploratory factor analysis was implemented in FACTOR (version 10.10.01, [65]). Since the data was ordinal, the analysis procedure based on polychoric correlations that was outlined by Baglin [66] was used as a starting point. Firstly, a parallel analysis based on minimum rank factor analysis was performed on bootstrapped polychoric correlation matrices with 1000 permutations, using the 95% quantile criterion threshold to determine the number of factors to extract [67]. The extracted factors were then rotated using the promin method of oblique rotation [68].

Rather than observed item responses loading neatly onto distinct factors, bifactor models represent observed responses to each item as a latent combination of the influences of a general factor and a group factor. Thus, bifactor models are used to help evaluate the plausibility of overall scores (i.e. unidimensionality) and subscale scores (i.e. multidimensionality, [69]). The follow-up exploratory bifactor analysis was also conducted using FACTOR, with a robust unweighted least squares (ULS) method of extraction and a promin rotation [70]. Confidence intervals for estimated parameters were calculated using the bias-corrected and accelerated percentile method based on a bootstrapped distribution of 1000 samples.

The Rasch analysis was conducted using the RUMM2030+ package [71] to further test the assumption of unidimensionality in the FTD-SS responses in Sample 1. Chi-square item-trait interaction statistics were calculated to define the overall fit of the unidimensional model for the FTD-SS, where a non-significant chi-square probability value indicates that the hierarchical ordering of the FTD-SS items is consistent across all levels of the underlying trait (see [72]). Additional evidence of unidimensionality was investigated using a principal components analysis/t-test protocol. A principle components analysis was conducted on the residuals produced from Rasch measurement model predictions, with varimax rotation. Item residual loadings were used to identify two potential dimensions, from which two sets of person measures were derived. A series of t-tests were then conducted for each pair of person measures to assess their equivalency. Evidence of unidimensionality is found when less than 5% of the t-tests are significant, or if the lower bound of the binomial 95% confidence interval overlaps 5%.

Confirmatory factor analyses were conducted on FTD-SS responses in Sample 2 using the "lavaan" package (version 0.6–6, [73]) in R (version 3.6.3, [74]). The models that were tested were derived from the results of the exploratory analyses in Sample 1. Robust ULS estimation (mean and variance adjusted) based on polychoric correlations was used to maintain consistency with the exploratory analyses. Ordinal coefficient alphas ($\alpha$) were calculated to estimate test consistency in both samples using the "psych" package (version 1.9.11, [75]) in R [74].

Finally, convergent validity of responses on the FTD-SS were investigated using multiple regression analyses. In particular, two multiple regression analyses were conducted to determine which scores from the four O-LIFE scales (Regression Model 1) and nine SPQ subscales (Regression Model 2) were the most important predictors of FTD-SS total scores, as well as whether these relationships were independent of IPIP Extraversion and Neuroticism. The influence of demographic variables on the FTD-SS total scores was also investigated using a series of one-way between-subjects analyses of variance (ANOVAs), with follow-up pairwise comparisons conducted using Tukey's Honestly Significant Difference (HSD) post-hoc tests (for variables with more than two levels). Welch's ANOVAs and Games-Howell post-hoc tests were used instead for variables showing evidence of unequal variances. Effect sizes were estimated using omega squared ($\omega^2$) for each main effect and Hedge's $g_s$ for significant pairwise comparisons. The influence of age was investigated using a Pearson's product-moment correlation.

The ANOVAs and Tukey's HSD post-hoc tests were performed using the base "stats" package in R [74]. Welch's ANOVAs and Games-Howell post-hoc tests were conducted using the "rstatix" package (version 0.7.0, [76]). The calculation of effect sizes was performed using the "effectsize" package (version 0.4.4, [77]). All bivariate and partial correlation analyses were performed using the "psych" package [75].

## Results

Sample 1 consisted of a total of 348 survey responses that were collected from 11/07/2019 to 20/11/2019. Of these, 23 did not meet inclusion criteria. An additional response was identified as a duplicate (i.e. one person completed the survey twice). Thus, the final sample size was 324. Sample 2 consisted of a total of 642 survey responses that were collected from 21/08/2019 to 24/06/2020. Of these, 23 did not meet inclusion criteria, two did not complete the FTD-SS, and seven were identified as duplicates. Thus, the final size of Sample 2 was 610, with 366 participants recruited via REP, 226 participants recruited via Prolific and 18 participants recruited via snowballing. Descriptive statistics pertaining to demographics, FTD-SS and schizotypy scale scores for both samples were included in Table 1.

No differences in age, employment status, past mental health diagnoses and family histories of schizophrenia-related disorders were found between Sample 1 and Sample 2 (see Table 1). However, differences were found in reported gender, student status, highest level of completed education and country of birth. These demographic differences were mostly attributable to the additional avenues used to recruit Sample 2. Nevertheless, a higher proportion of males and full-time students were recruited from REP in Sample 1 compared to those recruited from REP in Sample 2 (data not shown).

### Exploratory dimension reduction analyses

Responses to the FTD-SS items in Sample 1 were positively skewed. In particular, the third and fourth response categories had low rates of endorsement across all 29 items (ranging between 2.8% and 11.4% for the third category and between 0.3% and 3.4% for the fourth category). The modal response category was 2 for six of the items and 1 for the remaining 23

**Table 1. Descriptive statistics for demographic variables and survey measures across both samples.**

| | Sample 1 (n = 324) | Sample 2 (n = 610) | Between-Group Comparison |
|---|---|---|---|
| Age (years)[a] | 30.63 (10.70, 18–61) | 31.27 (10.60, 18–71) | $W = 94140$, $p = 0.26$ |
| Gender | | | |
| Male (%) | 26.54 | 34.59 | $\chi^2(1) = 5.95$, $p = 0.01$ |
| Female (%) | 72.84 | 65.08 | $\chi^2(1) = 5.48$, $p = 0.02$ |
| Non-binary (%) | 0.62 | 0.33 | $OR = 1.89$, $p = 0.61$ |
| Missing (%) | 0.00 | 0.00 | n/a |
| Country of Birth | | | |
| Australia (%) | 81.79 | 72.95 | $\chi^2(1) = 8.59$, $p = 0.003$ |
| Other (%) | 17.90 | 26.89 | $\chi^2(1) = 8.94$, $p = 0.003$ |
| Missing (%) | 0.31 | 0.16 | $OR = 1.88$, $p = 1.00$ |
| Highest Level of Completed Education | | | |
| Primary School (%) | 0.62 | 0.98 | $OR = 0.63$, $p = 0.72$ |
| Secondary School (%) | 38.58 | 28.20 | $\chi^2(1) = 10.05$, $p = 0.002$ |
| Technical or vocational training (%) | 38.89 | 31.80 | $\chi^2(1) = 4.41$, $p = 0.04$ |
| Undergraduate university degree (%) | 17.90 | 27.70 | $\chi^2(1) = 10.53$, $p = 0.001$ |
| Postgraduate university degree (%) | 4.01 | 11.31 | $\chi^2(1) = 13.18$, $p < 0.001$ |
| Missing (%) | 0.00 | 0.00 | n/a |
| Student Status | | | |
| Full-time (%) | 56.17 | 39.01 | $\chi^2(1) = 24.48$, $p < 0.001$ |
| Part-time (%) | 43.83 | 35.08 | $\chi^2(1) = 6.50$, $p = 0.01$ |
| Not a student (%) | 0.00 | 25.90 | $OR = 0.00$, $p < 0.001$ |
| Missing (%) | 0.00 | 0.00 | n/a |
| Employment Status[b] | | | |
| Unemployed (%) | 14.51 | 18.69 | $\chi^2(1) = 2.00$, $p = 0.16$ |
| Casual or part-time (%) | 38.89 | 35.74 | $\chi^2(1) = 1.27$, $p = 0.26$ |
| Full-time (%) | 34.26 | 35.25 | $\chi^2(1) = 0.01$, $p = 0.92$ |
| Retired, self-employed or other (%) | 16.05 | 14.92 | $\chi^2(1) = 0.00$, $p = 1.00$ |
| Missing (%) | 0.31 | 0.00 | n/a |
| Past Mental Health Diagnosis | | | |
| Yes (%) | 16.98 | 13.61 | $\chi^2(1) = 1.65$, $p = 0.20$ |
| No (%) | 83.02 | 86.39 | $\chi^2(1) = 1.65$, $p = 0.20$ |
| Missing (%) | 0.00 | 0.00 | n/a |
| Immediate Family Member with a Schizophrenia-Related Disorder | | | |
| Yes (%) | 5.56 | 6.23 | $\chi^2(1) = 0.07$, $p = 0.79$ |
| No (%) | 87.96 | 87.87 | $\chi^2(1) = 0.00$, $p = 1.00$ |
| Unsure (%) | 6.48 | 5.90 | $\chi^2(1) = 0.04$, $p = 0.83$ |
| Missing (%) | 0.00 | 0.00 | n/a |
| FTD-SS Scores | | | |
| Total (29 Items)[a] | 43.85 (10.76, 29–86) | 44.57 (10.97, 29–86) | $W = 94509$, $p = 0.27$ |
| Missing (%) | 0.00 | 0.00 | n/a |
| O-LIFE Scores | | | |
| Cognitive Disorganization[a] | 10.77 (6.23, 0–24) | 10.34 (6.29, 0–24) | $W = 101012$, $p = 0.31$ |
| Unusual Experiences[a] | 7.58 (6.65, 0–29) | 6.41 (5.82, 0–27) | $W = 105803$, $p = 0.02$ |
| Impulsive Nonconformity[a] | 7.57 (3.79, 0–19) | 6.84 (3.41, 0–18) | $W = 107743$, $p = 0.006$ |
| Introvertive Anhedonia[a] | 7.33 (4.78, 0–23) | 7.72 (4.89, 0–23) | $W = 92657$, $p = 0.25$ |
| Missing (%) | 0.93 | 0.82 | $OR = 1.15$, $p = 1.00$ |
| SPQ Scores | | | |

(Continued)

**Table 1.** (Continued)

| | Sample 1 (*n* = 324) | Sample 2 (*n* = 610) | Between-Group Comparison |
|---|---|---|---|
| Total[a] | 21.30 (13.98, 0–66) | 21.72 (13.71, 0–64) | *W* = 95951, *p* = 0.65 |
| Ideas of Reference[a] | 2.64 (2.51, 0–9) | 2.50 (2.40, 0–9) | *W* = 100443, *p* = 0.48 |
| Excessive Social Anxiety[a] | 3.86 (2.76, 0–8) | 3.96 (2.63, 0–8) | *W* = 95500, *p* = 0.56 |
| Odd Beliefs or Magical Thinking[a] | 1.46 (1.71, 0–7) | 1.26 (1.62, 0–7) | *W* = 103977, *p* = 0.09 |
| Unusual Perceptual Experiences[a] | 1.75 (2.03, 0–8) | 1.69 (1.86, 0–9) | *W* = 97126, *p* = 0.87 |
| Odd or Eccentric Behaviour[a] | 1.73 (2.05, 0–7) | 1.80 (2.11, 0–7) | *W* = 96600, *p* = 0.76 |
| No Close Friends[a] | 2.71 (2.53, 0–9) | 3.22 (2.67, 0–9) | *W* = 86927, *p* = 0.005 |
| Odd Speech[a] | 2.66 (2.24, 0–9) | 2.61 (2.36, 0–9) | *W* = 100259, *p* = 0.51 |
| Constricted Affect[a] | 2.05 (1.86, 0–8) | 2.20 (2.00, 0–8) | *W* = 94663, *p* = 0.42 |
| Suspiciousness[a] | 2.44 (2.27, 0–8) | 2.48 (2.26, 0–8) | *W* = 92698, *p* = 0.71 |
| Missing (%) | 0.62 | 0.49 | OR = 1.28, *p* = 1.00 |
| IPIP Scores | | | |
| Neuroticism[a] | 66.13 (15.11, 28–105) | 66.19 (15.21, 30–106) | *W* = 97850, *p* = 0.94 |
| Extraversion[a] | 75.94 (12.64, 40–106) | 74.40 (13.11, 35–107) | *W* = 103359, *p* = 0.14 |
| Missing (%) | 0.62 | 0.66 | OR = 0.94, *p* = 1.00 |

*Note.* Two-sample z-tests were performed to compare sample proportions, with Fisher's exact test used with counts < 5. Age and scale scores were compared using the Wilcoxon rank sum test because some distributions were skewed. All comparisons were two-tailed without correction for multiple comparisons.

[a] Mean (*SD*, range).

[b] Participants could select more than one response. Participants who selected more than one response (Sample 1: *n* = 14, Sample 2: *n* = 28; $\chi^2[1]$ = 0.00, *p* = 0.98) were excluded from between-groups comparisons.

Abbreviations: FTD-SS–Formal Thought Disorder–Self Scale; IPIP–International Personality Item Pool; O-LIFE–Oxford-Liverpool Inventory of Feelings and Experiences; SPQ–Schizotypal Personality Questionnaire

items. The items with the lowest response variances were 7, 22 and 25, where ~80% of the sample endorsed the first response category. Mardia's tests of multivariate normality showed a significant deviation in kurtosis (*p* < 0.001) but not skewness (*p* = 1.00). Means, standard deviations, and skewness and kurtosis statistics for the sampled item responses are presented in Table 2.

The Kaiser-Meyer-Olkin (KMO) index of sampling adequacy was 0.77 ($CI_{95\%}$ [0.630, 0.770]) and Bartlett's test of sphericity was significant ($\chi^2_{[406]}$ = 3597.80, *p* < 0.001), suggesting that the inter-item polychoric correlation matrix produced from the sampled data was suitable for factor analysis. The parallel analysis indicated the presence of three factors, which cumulatively explained 56.39% of the variance in the initial correlation matrix and 60.75% of the variance in the reduced correlation matrix. However, despite the inter-item polychoric correlation matrix being positive definite, the factor solution was inadmissible, with communalities of 1.00 for six items (i.e. Heywood cases; items 1, 15, 16, 23, 24 and 29). Item communalities improved when the factor analysis was repeated with the use of the ULS method of estimation, resulting in a solution with three highly correlated factors (*r* ≥ 0.56). However, simple structure was still not achieved, with cross-loadings above 0.30 for eight items. Moreover, three items (items 1, 9 and 10) demonstrated rotated factor loadings that exceeded 1.00, although this can occur with oblique rotation when the factors are strongly correlated with one another [78].

The exploratory bifactor analysis [70] was then conducted with three group factors to evaluate the plausibility of a multidimensional model of the FTD-SS responses [69]. ULS was chosen as the method of estimation for the bifactor analysis because it produced more appropriate communalities, although the pattern of factor loadings was similar when minimum rank

**Table 2. Descriptive statistics for item responses on the FTD-SS in Sample 1, as well as communalities, estimated factor loadings and inter-factor correlations for the final 20-item three-factor solution.**

| Item | *M* | *SD* | Skew. | Kurt. | Comm. | F1 | F2 | F3 |
|------|-----|------|-------|-------|-------|-----|-----|-----|
| 1 | 1.74 | 0.65 | 0.39 | -0.41 | 0.72 | **0.92** | -0.17 | -0.01 |
| 2 | 1.76 | 0.65 | 0.49 | 0.20 | 0.61 | **0.82** | -0.18 | 0.05 |
| 3 | 1.60 | 0.63 | 0.64 | -0.18 | 0.65 | **0.83** | -0.05 | -0.00 |
| 4 | 1.76 | 0.71 | 0.65 | 0.12 | 0.49 | **0.75** | 0.10 | -0.18 |
| 5 | 1.79 | 0.63 | 0.43 | 0.35 | 0.46 | **0.69** | 0.02 | -0.03 |
| 6 | 1.45 | 0.67 | 1.51 | 2.08 | 0.53 | **0.61** | 0.21 | -0.01 |
| 7 | 1.21 | 0.51 | 2.60 | 6.45 | | | | |
| 8 | 1.41 | 0.63 | 1.50 | 2.01 | 0.57 | 0.24 | **0.59** | 0.04 |
| 9 | 1.76 | 0.76 | 0.85 | 0.45 | 0.64 | -0.04 | **0.87** | -0.13 |
| 10 | 1.66 | 0.75 | 1.00 | 0.57 | 0.79 | 0.02 | **0.97** | -0.18 |
| 11 | 1.68 | 0.83 | 1.15 | 0.76 | 0.53 | 0.10 | **0.56** | 0.17 |
| 12 | 1.62 | 0.80 | 1.22 | 0.95 | 0.49 | 0.22 | 0.12 | **0.46** |
| 13 | 1.45 | 0.69 | 1.51 | 1.80 | 0.50 | 0.23 | -0.01 | **0.54** |
| 14 | 1.27 | 0.58 | 2.45 | 6.26 | | | | |
| 15 | 1.30 | 0.58 | 2.06 | 4.42 | 0.56 | **-0.33** | 0.14 | **0.82** |
| 16 | 1.39 | 0.65 | 1.75 | 3.01 | 0.67 | 0.02 | 0.08 | **0.76** |
| 17 | 1.43 | 0.64 | 1.44 | 1.75 | 0.34 | -0.18 | -0.16 | **0.74** |
| 18 | 1.50 | 0.68 | 1.33 | 1.52 | 0.53 | 0.08 | -0.01 | **0.69** |
| 19 | 1.31 | 0.55 | 1.84 | 3.55 | 0.46 | -0.14 | 0.06 | **0.72** |
| 20 | 1.55 | 0.68 | 1.14 | 1.12 | 0.44 | -0.03 | **0.47** | 0.30 |
| 21 | 1.57 | 0.71 | 1.06 | 0.56 | | | | |
| 22 | 1.25 | 0.51 | 2.07 | 4.19 | | | | |
| 23 | 1.56 | 0.77 | 1.32 | 1.14 | 0.42 | 0.10 | -0.12 | **0.64** |
| 24 | 1.57 | 0.69 | 1.05 | 0.66 | | | | |
| 25 | 1.25 | 0.51 | 2.12 | 0.44 | | | | |
| 26 | 1.71 | 0.73 | 0.85 | 0.51 | | | | |
| 27 | 1.37 | 0.65 | 1.85 | 3.26 | 0.53 | 0.24 | -0.07 | **0.59** |
| 28 | 1.50 | 0.66 | 1.18 | 0.98 | | | | |
| 29 | 1.47 | 0.74 | 1.73 | 2.69 | | | | |
| | | | | Inter-Factor Correlations | | | | |
| | | | | | F1 | - | | |
| | | | | | F2 | 0.46 | - | |
| | | | | | F3 | 0.62 | 0.56 | - |

*Note.* Factor loadings > 0.30 are presented in bold. Communalities and factor loadings are omitted for nine FTD-SS items that were excluded from the exploratory factor analysis based on weak factor loadings in the bifactor model (see Table A in S1 File). Factor labels: F1—Difficulty with Maintaining the Topic of Conversation; F2—Difficulty with Initiating and Sustaining Speech; F3—Odd Speech.

estimation was performed instead. The results were somewhat mixed (see Table A in S1 File). In support of essential unidimensionality, seven items failed to load significantly on any of the group factors, and closeness-to-unidimensionality indices for each of the items indicated that most of the item-variances were accounted for by the general factor (see Table B in S1 File). Yet, two items failed to load significantly on the general factor, and the mean factor loadings for each of the group factors were comparable to that for the general factor (see Table A in S1 File). Moreover, although the three closeness-to-unidimensionality indices approached the threshold for unidimensionality of the overall model, only one of these indices actually reached

this threshold (see Table B in S1 File). Similarly, whilst the model fit statistics from the Rasch analysis suggested that the unidimensional model was a reasonable fit ($\chi^2$ = 167.84, $p$ = 0.001; PRI = 0.873; IRI = 0.919; PSI = 0.877; ISI = 0.892), the chi-squared test was marginally significant after Bonferroni correction for 29 comparisons. Furthermore, 9.6% of the equivalency t-tests were significant ($CI_{95\%}$ [7.2%, 11.9%]), exceeding the 5% threshold for unidimensionality.

Next, the results of the exploratory bifactor analysis were used to identify and exclude potentially problematic items (see S1 File). Exploratory factor analyses with ULS estimation were systematically repeated after removing the two FTD-SS items that did not show significant factor loadings on the general factor and the seven items that did not show significant loadings on any of the three group factors in the bifactor model (see Table C in S1 File). However, simple structure was only achieved after removing all nine of these items (see Table 2), resulting in three correlated factors that explained 60.90% of the variance in the polychoric correlation matrix for the remaining 20 items. Fit indices suggested that this three-factor model described the data well (see Table C in S1 File). Notably, the removal of three different items that showed high inter-item correlations alternatively improved the evidence for unidimensionality amongst the remaining 26 items. However, evidence of collinearity and item redundancy that would support the removal of these items was ultimately weak (see S1 File).

Finally, scores derived from the 20-item three-factor solution were evaluated for their factor determinacy and reliability [79]. Refined factor score estimates calculated using a linear regression-based approach (see [79, 80]) for the three factors were strongly correlated with one another (see Table D in S1 File), closely replicating the inter-factor correlations themselves (see Table 2). However, simple summed scores (i.e. the unweighted sum of observed item responses) for the three factors also replicated these inter-factor correlations (see Table E in S1 File) and were very highly correlated with the refined factor score estimates (see Table F in S1 File). The use of total scores was still supported by most of the closeness-to-unidimensionality indices for the 20-item three-factor model (see S1 File). Coefficient alphas calculated based on polychoric correlations indicated a high degree of consistency for each of the three factors (F1: $\alpha$ = 0.88; F2: $\alpha$ = 0.86; F3: $\alpha$ = 0.89), as well as across all 20 items ($\alpha$ = 0.92).

The 20-item version of the FTD-SS is referred to hereon in as the Revised Formal Thought Disorder–Self-Report Scale (FTD-SS-R). The first group factor encompasses a breakdown in the direction or purpose of speech and was labelled 'Difficulty with Maintaining the Topic of Conversation'. The second group factor represents speech that is constrained or deficient and was labelled 'Difficulty with Initiating and Sustaining Speech'. The third group factor encompasses the peculiar and idiosyncratic elements of speech, and so was labelled 'Odd Speech'.

## Confirmatory factor analysis

FTD-SS item responses in Sample 2 were positively skewed, similarly to those of Sample 1. Confirmatory factor analyses were used to assess the fit of a unidimensional model of responses to all 29 FTD-SS items in this second sample (see Fig 1), as well as Barrera et al.'s [56] correlated three factors model (see Fig 2). The fit of a unidimensional model of responses to the 20 items retained in the final exploratory factor solution in Sample 1 (see Fig 3), as well as the final correlated three factors model of responses to these 20 items (see Fig 4), were also assessed in Sample 2. As seen in Table 3, robust fit indices were superior for the correlated three factors models than the unidimensional models, with the 20-item three factors model describing the data most accurately. The coefficient alphas produced from this model of FTD-SS-R responses in Sample 2 were almost identical to those in Sample 1 (F1: $\alpha$ = 0.86; F2: $\alpha$ = 0.85; F3: $\alpha$ = 0.88; total: $\alpha$ = 0.92). The factor loadings and inter-factor correlations produced for this model have been presented in Table G in S1 File.

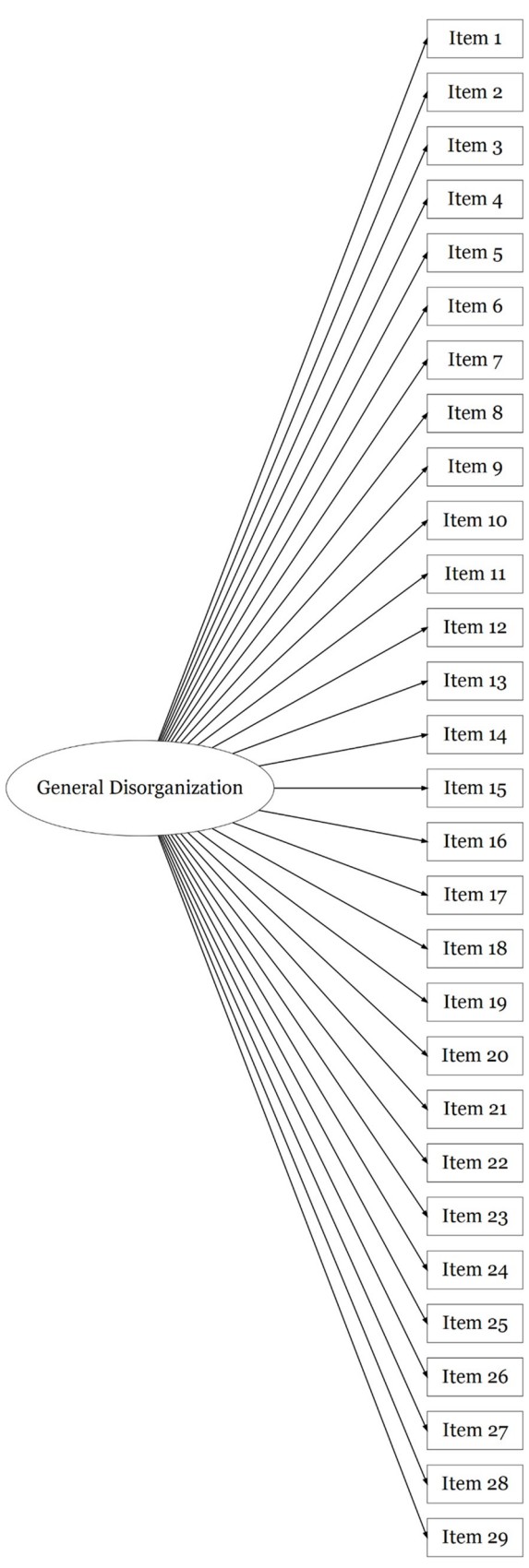

**Fig 1. A diagrammatic representation of the unidimensional model of FTD-SS item-responses tested using a confirmatory factor analysis in Sample 2, where the variation in all 29 item-responses is explained by a single factor (i.e. 29-item unidimensional model).**

## Convergent relationships with schizotypy and the influence of demographics

No evidence of significant differences in FTD-SS total scores were found between Sample 1 and Sample 2 (see Table 1). Thus, the two participant groups were combined for all subsequent analyses ($N$ = 934). Moreover, for the following analyses, total scores were calculated from responses to the 20-item FTD-SS-R that were included in the best-fitting correlated three factors model. Lastly, to produce regression models that demonstrated linearity, homoscedasticity and normally distributed residuals, the FTD-SS-R total scores were transformed using a negative reciprocal transformation. However, the results were similar when untransformed summed scores were analysed instead.

Associations between the transformed FTD-SS-R scores and the other measures of schizotypy and personality were all highly significant (see Table 5). Individuals who obtained higher FTD-SS-R scores also tended to obtain higher scores on all of the other schizotypy scales, as well as on the neuroticism scale, but tended to obtain lower extraversion scores (i.e. they tended to be more introverted). Scores on the O-LIFE Cognitive Disorganization scale and the SPQ Odd Speech subscale were the strongest correlates of FTD-SS-R total scores and were most important independent predictors of FTD-SS-R total scores in the multiple regression analyses (in Models 1 and 2, respectively; see Table 4). Notably, however, both models only explained around half the variance in FTD-SS-R total scores, with 9% of the variance in FTD-SS-R total scores being explained by scores on the O-LIFE Cognitive Disorganization scale independently of scores on the other O-LIFE and IPIP scales, and 16% of the variance in FTD-SS-R total scores was explained by scores on the SPQ Odd Speech subscale independently of scores on the other SPQ subscales and IPIP scales.

The results of these regression analyses were almost identical when total scores were calculated using all of the original 29 items. Specifically, the effect sizes of all zero-order correlations, partial correlations, standardized regression coefficients, and $R^2$ values for the 29-item FTD-SS total scores in both regression models were within 0.03 of those for the 20-item FTD-SS-R total scores. In addition, the convergent patterns of bivariate correlations were generally similar for summed subscale scores calculated for the three FTD-SS-R factors (see Table H in S1 File). Nevertheless, compared to the other FTD-SS-R subscale scores, Difficulty Initiating and Sustaining Conversation scores tended to be more strongly associated with scores on the negative schizotypy subscales (e.g. O-LIFE Introvertive Anhedonia, SPQ Constricted Affect, etc.), as well as levels of introversion, and more weakly associated with scores on most of the positive schizotypy subscales (e.g. O-LIFE Unusual Experiences, SPQ Unusual Perceptual Experiences, etc.).

Regarding the effects of the demographic variables, transformed FTD-SS-R total scores tended to be greater in participants who were younger ($r$ = -0.23, $n$ = 933, $p < 0.001$) and were greater, on average, in males compared to females (see Table 5). Significant effects were also found for level of completed education and current student status, as well as employment status, personal history of mental illness and a family history of schizophrenia-related disorders. Mean transformed FTD-SS-R total scores were lower amongst people who completed a postgraduate degree relative to people who had completed secondary school or technical training, lower amongst current students enrolled part-time relative to full-time students and non-students, lower amongst people employed full-time relative to those who were unemployed, and

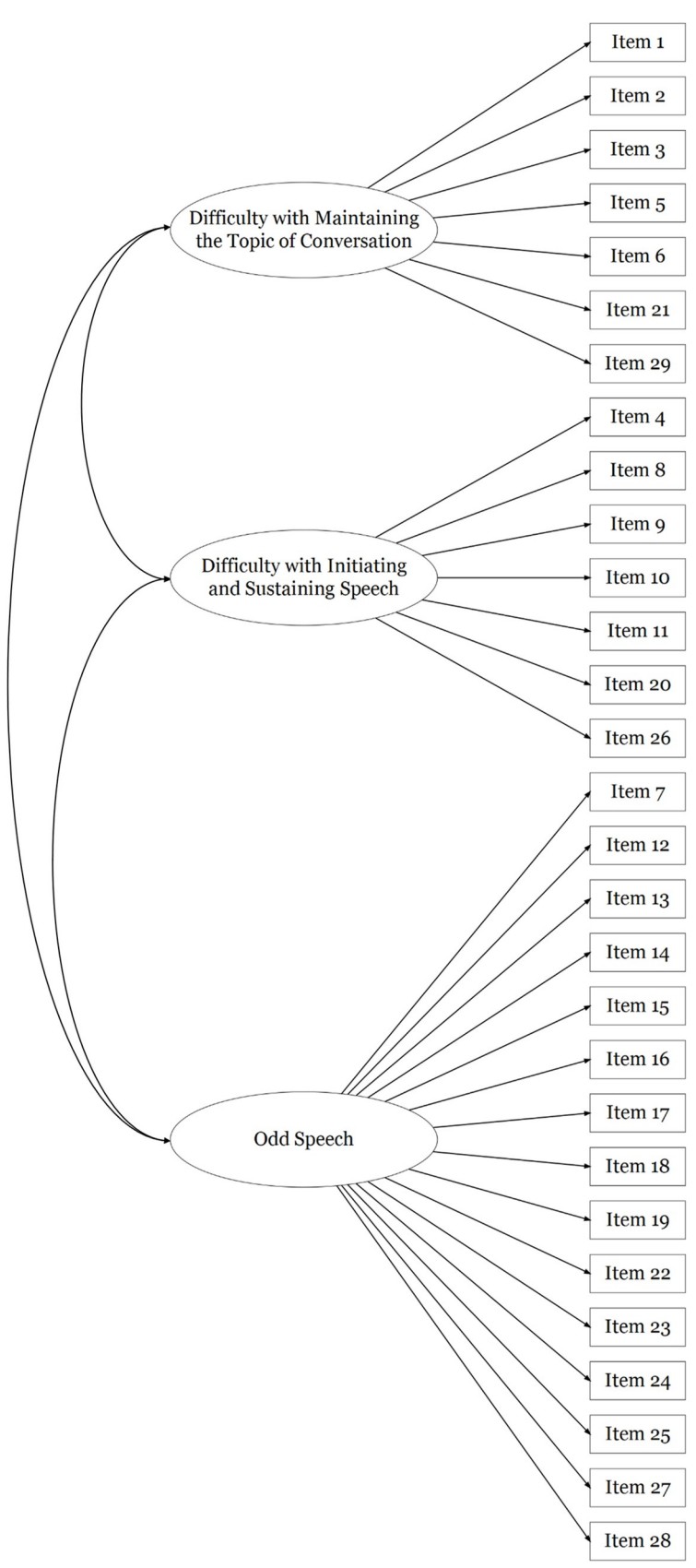

**Fig 2. A diagrammatic representation of the correlated three factors model of FTD-SS item-responses that was reported by Barrera et al.** [56] and tested using a confirmatory factor analysis in Sample 2, where the variation in all 29 item-responses is explained by three correlated factors (i.e. Barrera et al.'s 29-item correlated three factors model).

lower amongst those who had received a mental health diagnosis in the past. It should be emphasised that the sizes of all these effects were generally very weak. Again, these results were very similar when untransformed FTD-SS-R totals were analysed instead of the transformed scores. They were also similar when total scores from the original 29-item FTD-SS were analysed (see Table I in S1 File).

## Discussion

The aim of the current study was to investigate the dimensionality and convergent validity of responses to Barrera et al.'s [56] Formal Thought Disorder–Self-Report Scale (FTD-SS) obtained in two non-clinical samples. Despite some indications of unidimensionality, exploratory dimension-reduction analyses in the first sample ultimately supported the presence of three correlated factors. However, an acceptable factor solution was only found after the removal of nine FTD-SS items, resulting in a revised 20-item version of the scale (FTD-SS-R). Confirmatory factor analyses in the second sample supported these exploratory analyses. In particular, the correlated three factors model of responses to the FTD-SS-R provided a superior fit relative to unidimensional models of both the FTD-SS-R and the original 29-item FTD-SS. This model was also a better fit than Barrera et al.'s [56] correlated three factors model of responses to the 29-item FTD-SS. Finally, FTD-SS-R scores were found to be correlated appropriately with other schizotypy measures, and particularly strongly with subscales that include items designed to assess disorganised or constrained thought and speech (DCTS).

The main difference between the current findings and those of Barrera et al. [56] was the need to remove items from the original FTD-SS in this study. The factor solutions were otherwise very similar, with both studies indicating the presence of three correlated factors. Several methodological considerations may explain this discrepancy. Notably, in the current study, the initial exploratory factor analysis of responses to all 29 items produced an inadmissible three-factor solution with Heywood cases. Heywood cases can be a sign of model misspecification, inflated sampling error, or estimates of parameters in the population that lie close to the upper boundary [82]. Item responses that exhibit redundancy or very low variance (i.e. extreme response distributions) might also produce these cases.

In terms of the influence of the sample and sampling error on the discrepant findings, it is notable that the means and standard deviations for each of the item responses sampled by Barrera et al. [56] were larger than those sampled here, even though these participant samples seem comparable in size and demography. However, Barrera et al. conducted a principle components analysis, possibly based on a Pearson's inter-item correlation matrix (although this was not specified). Whilst polychoric correlation matrices are more appropriate for analyses of ordinal data [66], large sample sizes are required to overcome the increased sampling error associated with these correlations. This is particularly so when the sampled item responses are skewed or exhibit constrained variances, or when the measure has a low number of item-response categories or a large number of items [82]. Although there is a chance that sampling error played a part in the current findings, simulated data on a four-point ordinal scale with non-symmetrical distributions indicate that exploratory factor analyses with a sample of 350 responses should demonstrate a good quality three-factor solution across 30 items [83]. Thus, sample size alone is not likely to account for the discrepant findings.

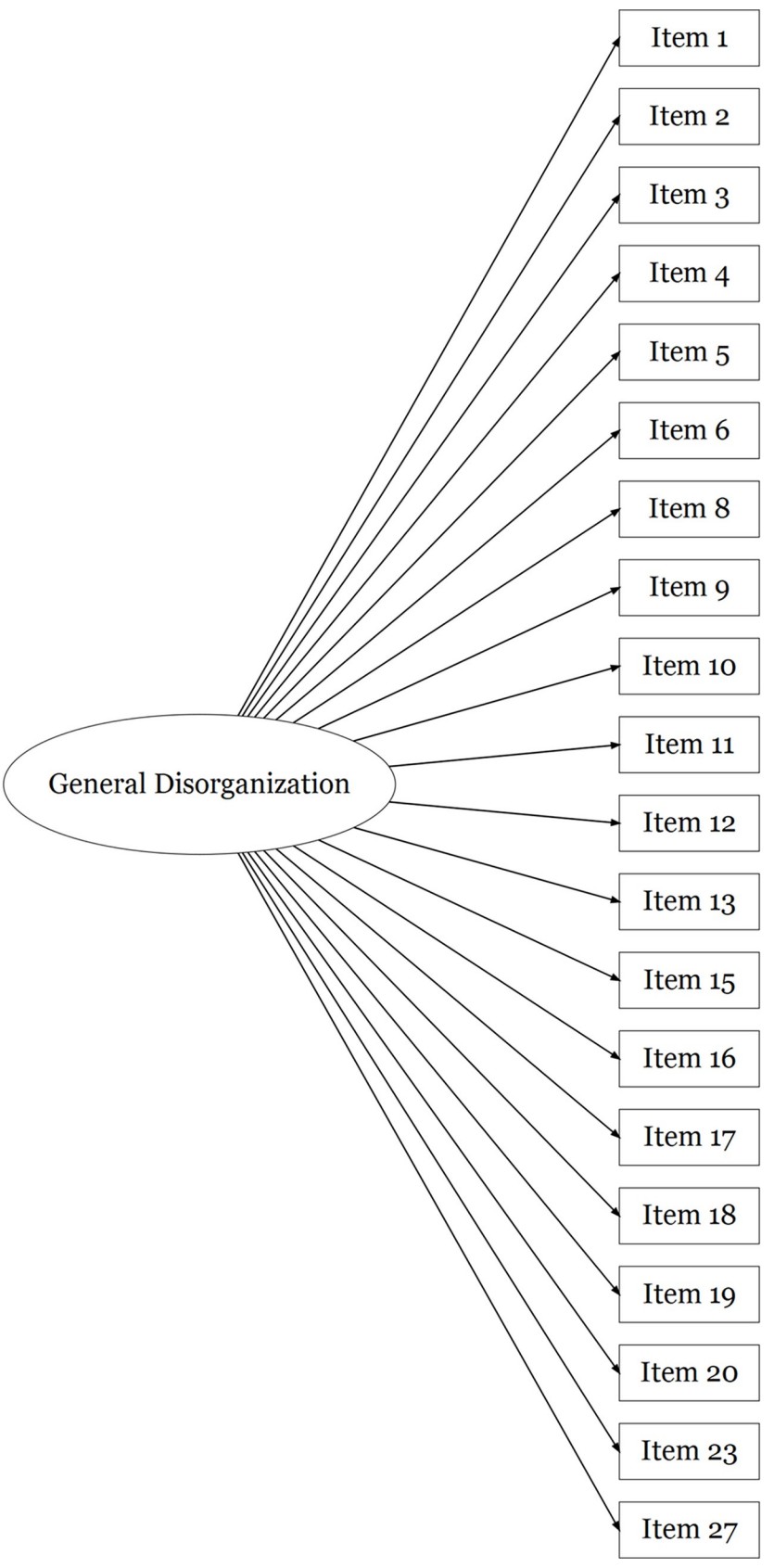

**Fig 3. A diagrammatic representation of the unidimensional model of FTD-SS-R item-responses tested using a confirmatory factor analysis in Sample 2, where a single factor explains the variation in item-responses for the 20 FTD-SS-R items that were retained in the final exploratory factor solution in Sample 1 (i.e. 20-item unidimensional model).**

As alluded to already, the item-response distributions tended to be positively skewed in both samples from the current study, with some items exhibiting low variance. Skewed distributions could characterise the continuum of psychopathology that encompasses schizotypy and psychosis when observed over the entire population [5, 20, 58]. However, some schizotypy measures in the past have been criticised for yielding heavily skewed data (e.g.[2, 84]), and this skew does call into question the relevance of many of the FTD-SS items to the majority of the general population. Given that nine items had to be removed in the current study before an adequate factor solution was found, a more thorough investigation of the performance of FTD-SS items is required in non-clinical samples who typically to report relatively infrequent experiences of DCTS. For example, follow-up Rasch analyses could be conducted to further support the current revisions to the FTD-SS.

It is not yet clear what variables influence FTD-SS scores amongst people without mental health disorders. Barrera et al. [56] found that FTD-SS scores tended to be greater in younger participants and amongst those reporting a personal history of mental health disorders, but did not differ between males and females or between those with and without a family history of mental health disorders. The current findings were similar, though FTD-SS-R total scores were also greater in males, and varied with education, and student and employment status. These relationships demonstrate some consistency with normative data from other schizotypy measures (e.g. [40]). However, the effects were quite small. Barrera et al. [56], in fact, collected a second sample of FTD-SS responses, reporting an average FTD-SS total score that was much closer to those of the current study. The authors speculated that the anonymous online completion of the FTD-SS may promote the disclosure of DCTS experiences when compared with the face-to-face data collection that occurred for their second sample. Yet, the current data was also collected anonymously and online. Thus, the important determinants of DCTS in schizotypy, as assessed by the FTD-SS, may not have been captured in the demographic variables assessed.

The underlying motivation behind the current study was the idea that a questionnaire dedicated to the assessment of DCTS in schizotypy could yield more sensitive measurements compared to global schizotypy measures. This idea stems from the heterogeneity that characterises DCTS in clinical samples [41–52]. Admittedly, the exact number and nature of DCTS dimensions have yet to be elucidated, even for observer-rated DCTS in clinical samples. Dimension reduction analyses have indicated anywhere between two and seven separate factors, depending on the particular measures used and aspects of DCTS sampled [41–52]. A negative DCTS factor has been identified reasonably consistently [41, 43–46, 48, 50], including with interview-based measures of subjective DCTS [46], and this factor is often associated more with the negative syndrome of schizophrenia than the positive or disorganized syndromes [43, 46, 47, 50, 85–88]. Nevertheless, other specific DCTS factors that have been reported seem to be somewhat poorly reproduced across studies [41, 43–47, 49–51], although there are examples of three-factor models (e.g [50]) with similarities to the three factors found in the current study. Moreover, unidimensional models in clinical samples tend to be a poor fit (e.g. [44]). Thus, the current results are consistent with some of the findings derived from clinical samples using objective measures of DCTS.

Whilst multidimensionality was supported in the current study, it is not yet clear whether the FTD-SS-R yields a more sensitive measure of DCTS than those afforded by global

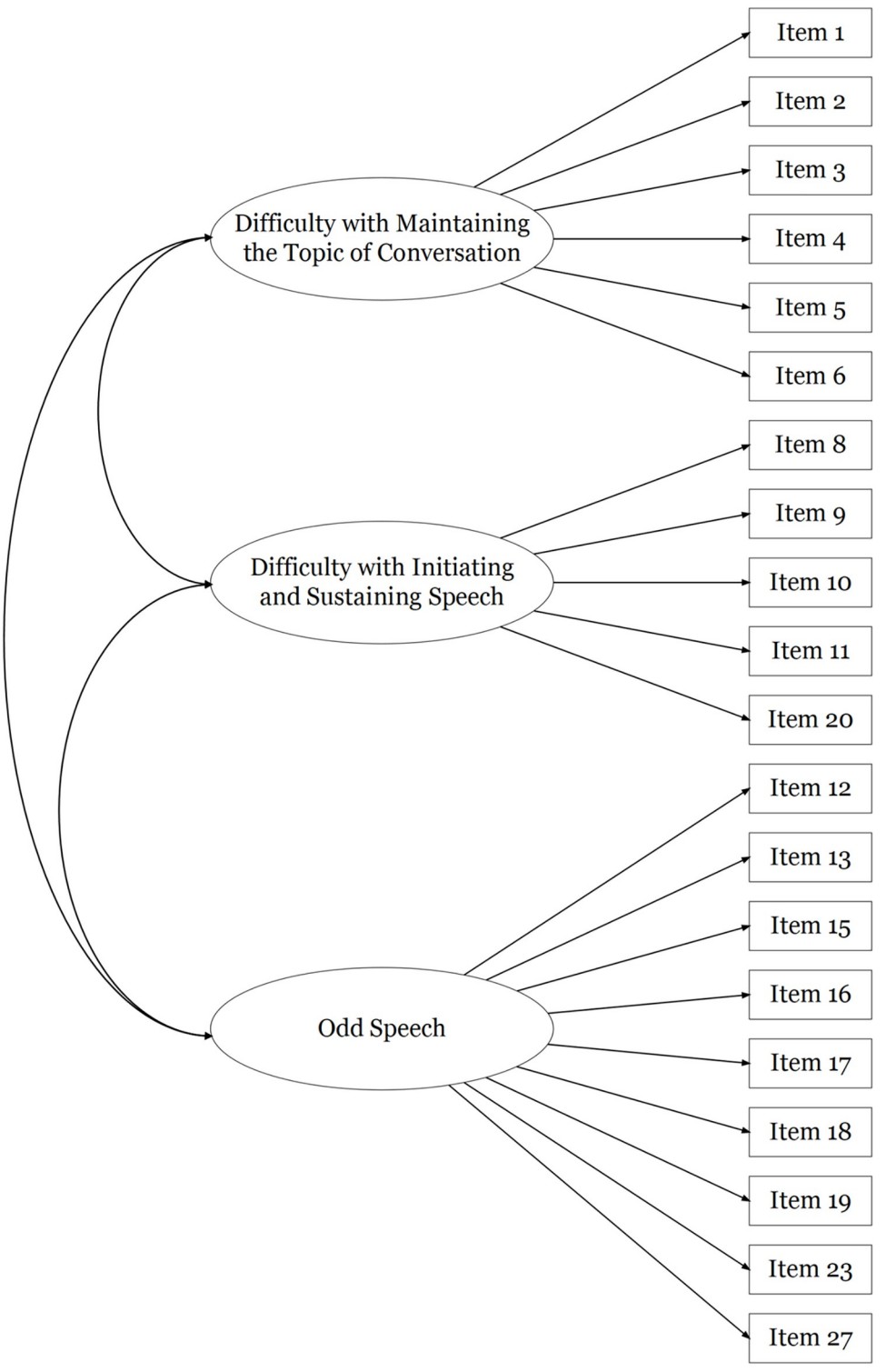

**Fig 4. A diagrammatic representation of the correlated three factors model of FTD-SS-R item-responses produced using an exploratory factor analysis in Sample 1 and tested using a confirmatory factor analysis in Sample 2, where the variation in item-responses for 20 of the FTD-SS-R items is accounted for by three correlated factors (i.e. 20-item correlated three factors model).**

**Table 3. Robust fit indices for the four models tested using confirmatory factor analyses (Sample 2).**

| | $\chi^2$ (df) | CFI | TLI | RMSEA | SRMR | Average Item $R^2$ |
|---|---|---|---|---|---|---|
| Unidimensional Model (29-Item FTD-SS) | 1342.58 (377) | 0.89 | 0.88 | 0.07 | 0.08 | 0.38 |
| Barrera et al.'s [56] Correlated Three Factors Model (29-Item FTD-SS) | 987.60 (374) | 0.93 | 0.92 | 0.05 | 0.07 | 0.44 |
| Unidimensional Model (20-Item FTD-SS-R) | 946.55 (170) | 0.86 | 0.84 | 0.09 | 0.09 | 0.38 |
| Correlated Three Factors Model (20-Item FTD-SS-R) | 436.45 (167) | 0.95 | 0.94 | 0.05 | 0.06 | 0.48 |

Abbreviations: CFI–Comparative Fit Index; TLI–Tucker-Lewis Index; RMSEA–Root Mean Square Error of Approximation; SRMR–Standardized Root Mean Square Residual

schizotypy questionnaires. Indeed, FTD-SS-R total scores showed appropriate convergence with other measures of disorganized schizotypy, and these relationships were not accounted for by neuroticism or extraversion. Moreover, the three factors found in this study were strongly correlated with one another, and closeness-to-unidimensionality indices generally supported the use of the total score. Yet, the combination of schizotypy, neuroticism and extraversion scores only managed to explain half of the variance in FTD-SS-R total scores. Notably,

**Table 4. Multiple regression analyses to investigate which O-LIFE scale scores (Model 1) and SPQ subscale scores (Model 2) represent the most important predictors of FTD-SS-R total scores (Transformed) independent of neuroticism and extraversion.**

| Predictors | Pearson's Correlation Coefficients ($r$) | Standardised Regression Coefficients ($\beta$) | Partial Correlation Coefficients ($r_{partial}$) |
|---|---|---|---|
| *Regression Model 1* | | | |
| O-LIFE Cognitive Disorganization | 0.64*** | 0.41*** | 0.30 |
| O-LIFE Unusual Experiences | 0.52*** | 0.21*** | 0.21 |
| O-LIFE Introvertive Anhedonia | 0.39*** | 0.09** | 0.09 |
| O-LIFE Impulsive Non-Conformity | 0.36*** | 0.07* | 0.08 |
| IPIP Neuroticism | 0.49*** | 0.00 | 0.00 |
| IPIP Extraversion | -0.37*** | -0.09* | -0.08 |
| | $R^2 = 0.47$, $F(6, 919) = 139.40$, $p < 0.001$ | | |
| *Regression Model 2* | | | |
| SPQ Ideas of Reference | 0.46*** | 0.03 | 0.03 |
| SPQ Excessive Social Anxiety | 0.48*** | 0.09** | 0.09 |
| SPQ Odd Beliefs or Magical Thinking | 0.21*** | 0.03 | 0.04 |
| SPQ Unusual Perceptual Experiences | 0.43*** | 0.12*** | 0.13 |
| SPQ Odd or Eccentric Behaviour | 0.45*** | 0.05 | 0.06 |
| SPQ No Close Friends | 0.46*** | 0.04 | 0.03 |
| SPQ Odd Speech | 0.67*** | 0.41*** | 0.40 |
| SPQ Constricted Affect | 0.48*** | 0.06 | 0.06 |
| SPQ Suspiciousness | 0.45*** | -0.04 | -0.04 |
| IPIP Neuroticism | 0.49*** | 0.14*** | 0.15 |
| IPIP Extraversion | -0.37*** | -0.06 | -0.06 |
| | $R^2 = 0.54$, $F(11, 916) = 99.06$, $p < 0.001$ | | |

Note.

* $p < .05$

** $p < .01$

*** $p < .001$. Probability values for Pearson's product-moment (zero-order) correlations were adjusted for multiple tests using the false discovery rate [81].

Abbreviations: FTD-SS-R–Formal Thought Disorder–Self Scale–Revised (20-Items); IPIP–International Personality Item Pool; O-LIFE–Oxford-Liverpool Inventory of Feelings and Experiences; SPQ–Schizotypal Personality Questionnaire

**Table 5. Comparisons of total scores (Transformed) from the 20-Item FTD-SS-R across demographic variables in the combined participant sample.**

| | $n$ | Mean (SD) | Significant Pairwise Comparisons[a] |
|---|---|---|---|
| Gender ($F_{[1, 928]} = 5.56$, $p = 0.02$, $\omega^2 = 0.00$, $CI_{90\%}$ [0.000, 0.018]) | | | |
| Male (M) | 297 | 32.33 (8.43) | M > F; $g = 0.17$, $CI_{95\%}$ (0.028, 0.304) |
| Female (F) | 633 | 30.88 (7.51) | |
| Country of Birth ($F_{[1, 930]} = 2.77$, $p = 0.10$, $\omega^2 = 0.00$, $CI_{95\%}$ [0.000, 0.012]) | | | |
| Australia | 710 | 31.58 (7.87) | |
| Other | 222 | 30.64 (7.72) | |
| Highest Level of Completed Education ($F_{[3, 922]} = 4.40$, $p = 0.01$, $\omega^2 = 0.01$, $CI_{95\%}$ [0.000, 0.025]) | | | |
| Secondary school (S) | 297 | 32.30 (8.10) | S > P; $g = 0.43$, $CI_{95\%}$ (0.188, 0.681) |
| Technical or vocational training (T) | 320 | 31.41 (8.06) | T > P; $g = 0.33$, $CI_{95\%}$ (0.060, 0.547) |
| Undergraduate university degree | 227 | 30.85 (7.09) | |
| Postgraduate university degree (P) | 82 | 29.35 (7.69) | |
| Student Status ($F_{[2, 931]} = 4.15$, $p = 0.02$, $\omega^2 = 0.01$, $CI_{95\%}$ [0.000, 0.020]) | | | |
| Full-time (Ft) | 420 | 31.87 (8.35) | Ft > Pt; $g = 0.17$, $CI_{95\%}$ (0.031, 0.314) |
| Part-time (Pt) | 356 | 30.29 (6.89) | Pt < N; $g = -0.24$, $CI_{95\%}$ (-0.426, -0.050) |
| Not a student (N) | 158 | 32.31 (8.20) | |
| Employment Status ($F_{[4, 241.07]} = 4.20$, $p = 0.003$, $\omega^2 = 0.05$, $CI_{95\%}$ [0.002, 0.100]) | | | |
| Unemployed (Un) | 147 | 32.76 (7.70) | Un > Ft; $g = 0.38$, $CI_{95\%}$ (0.182, 0.577) |
| Casual or part-time (Pt) | 322 | 31.48 (8.38) | Pt < Un; $g = -0.23$, $CI_{95\%}$ (-0.427, -0.036) |
| Full-time (Ft) | 312 | 30.24 (7.19) | |
| Self-employed | 50 | 30.86 (6.34) | |
| Other or mixed | 103 | 32.49 (8.38) | |
| Past Mental Health Diagnosis ($F_{[1, 201.15]} = 6.08$, $p = 0.01$, $\omega^2 = 0.02$, $CI_{95\%}$ [0.000, 0.081]) | | | |
| Yes (Y) | 138 | 32.51 (7.96) | Y > N; $g = 0.21$, $CI_{95\%}$ (0.029, 0.391) |
| No (N) | 796 | 31.14 (7.80) | |
| Immediate Family Member with a Schizophrenia-Related Disorder ($F_{[2, 931]} = 4.66$, $p = 0.01$, $\omega^2 = 0.01$, $CI_{95\%}$ [0.000, 0.022]) | | | |
| Yes | 56 | 32.89 (8.68) | |
| No (N) | 821 | 31.05 (7.64) | N < U; $g = -0.36$, $CI_{95\%}$ (-0.631, -0.093) |
| Unsure (U) | 57 | 34.07 (9.07) | |

*Note*. Only four participants identified their gender as non-binary and only eight participants completed primary school as their highest level of education. Thus, these responses were omitted from the current analyses. Transformed FTD-SS-R total scores (negative reciprocal) were the dependent variable in all analyses.

[a] Probability values for post-hoc tests (i.e. Tukey's HSD or Games-Howell tests) were adjusted for familywise error using the Bonferroni method, with an alpha criterion of $p_{adj} \leq .05$ (underlined: $0.05 < p_{adj} < .10$). Hedge's $g$ effect sizes provided for significant (or near-significant) pairwise comparisons. Post-hoc tests only performed for variables exhibiting significant main effects.

however, Sumner et al. [58] found relationships between measures of semantic dysfluency and FTD-SS scores that were not evident with O-LIFE cognitive disorganization scores. Since semantic dysfunction is a prominent hypothesis for the pathogenesis of DCTS [89], it is at least possible that FTD-SS-R scores exhibit incremental validity over more global schizotypy measures. This possibility deserves additional research.

Another important goal of future research will be to determine whether there are conditions wherein the FTD-SS-R group dimensions become more important, such as in samples that include people experiencing more severe forms of DCTS, or people with a family history of DCTS. Some dimensions of DCTS may vary continuously between clinical and non-clinical populations, but others may not. Alternatively, some dimensions of DCTS may not be accessible to self-report, and possible sources of divergence with objective DCTS measures should be

investigated. For instance, one study has demonstrated a wide range of self-reported DCTS scores obtained by people who did not have DCTS, at least ostensibly according to a clinician-rated scale [54]. It is possible that self-report questionnaires produce over-reporting in people without DCTS who are hypersensitive to relatively trivial difficulties with their communication. Relationships between self-report measures of disorganized schizotypy and neuroticism have been reported in the past [12, 13, 17], and some researchers have questioned whether self-report measures of disorganization represent anything other than a person's anxiety surrounding their own cognitive functioning [17, 19]. The current findings, however, suggest that the convergence between the FTD-SS-R and the other measures of disorganized schizotypy cannot sufficiently be explained by neuroticism.

Additionally, DCTS can be accompanied by poor insight. This is most pronounced amongst people experiencing acute and subacute episodes of psychosis or mania, some of whom apparently do not even recognise instances of gross impairment or impropriety that are present in their speech [90–92]. In these cases, subjective measures would be expected to produce under-reporting relative to objective measures [18, 54]. However, reduced insight is less common amongst individuals diagnosed with schizophrenia-related disorders whose symptomatology is mild and stable [93], and this relationship presumably extends to people without mental health diagnoses. Regardless, in some research contexts, variations in insight and other metacognitive abilities might preclude the use of the FTD-SS-R on its own without an objective measure of DCTS. The most pressing questions regarding the validity of subjective DCTS measures, such as the FTD-SS-R, is the degree to which they reflect objective manifestations of DCTS, as well as the implications of any divergence upon the pathogenesis, prognosis, and functional impact of DCTS throughout the entire psychopathological continuum [54].

One main limitation of the current study pertains to the demographic differences between the two current samples, as well as the representativeness of these samples. For example, the first sample consisted entirely of university students, and whilst the second sample included non-students, the participants were mostly well-educated. There might be reasons to expect greater levels of schizotypy amongst more disadvantaged groups [94]. Furthermore, although the assessed demographic variables had a relatively modest influence on FTD-SS-R total scores, the underlying dimensionality may still change across these variables. Thus, the inadequacy of the unidimensional model found in the second sample may have been related to these demographic differences. This should be assessed in future studies.

In conclusion, the current study was unable to entirely replicate Barrera et al.'s [56] correlated three factors model of FTD-SS responses in two non-clinical samples. After the removal of nine items, however, evidence did emerge in support of three correlated factors using exploratory factor analysis in the first sample, and this model was supported using confirmatory factor analysis in the second sample. Future research should investigate the performance of the FTD-SS items in more detail to confirm the suggested revisions. Finally, FTD-SS-R total scores exhibited evidence of convergent validity through their associations with other measures of schizotypy, and this convergence could not entirely be explained by neuroticism or extraversion.

## Supporting information

**S1 File. Supplmental analyses and considerations.**
(DOCX)

**S1 Dataset. FTD-SS Item responses and sample labels.**
(TXT)

## Author Contributions

**Conceptualization:** Philip J. Sumner.

**Data curation:** Philip J. Sumner.

**Formal analysis:** Philip J. Sumner, Denny Meyer, Fakir M. Amirul Islam.

**Funding acquisition:** Susan L. Rossell.

**Investigation:** Philip J. Sumner.

**Methodology:** Philip J. Sumner, Sean P. Carruthers.

**Project administration:** Philip J. Sumner.

**Resources:** Susan L. Rossell.

**Software:** Philip J. Sumner, Fakir M. Amirul Islam.

**Supervision:** Susan L. Rossell.

**Validation:** Philip J. Sumner.

**Visualization:** Philip J. Sumner.

**Writing – original draft:** Philip J. Sumner.

**Writing – review & editing:** Philip J. Sumner, Denny Meyer, Sean P. Carruthers, Fakir M. Amirul Islam, Susan L. Rossell.

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
