## [Decision Letter · Decision Letter 0]

3 Dec 2021

PONE-D-21-24412Assessing the dimensionality of scores derived from the Formal Thought Disorder Self-Report Scale in schizotypyPLOS ONE

Dear Dr. Sumner,

Thank you for submitting your manuscript to PLOS ONE. After careful consideration, we feel that it has merit but does not fully meet PLOS ONE’s publication criteria as it currently stands. Therefore, we invite you to submit a revised version of the manuscript that addresses the points raised during the review process.

plosone@plos.org. Please include the following items when submitting your revised manuscript:A rebuttal letter that responds to each point raised by the academic editor and reviewer(s). You should upload this letter as a separate file labeled 'Response to Reviewers'.A marked-up copy of your manuscript that highlights changes made to the original version. You should upload this as a separate file labeled 'Revised Manuscript with Track Changes'.An unmarked version of your revised paper without tracked changes. You should upload this as a separate file labeled 'Manuscript'.

We look forward to receiving your revised manuscript.

Kind regards,

Marco Innamorati

Academic Editor

PLOS ONE

Journal Requirements:

Reviewers' comments:

Reviewer's Responses to Questions

**Comments to the Author**

1. Is the manuscript technically sound, and do the data support the conclusions?

Reviewer #1: No

Reviewer #2: Yes

2. Has the statistical analysis been performed appropriately and rigorously? 

Reviewer #1: No

Reviewer #2: Yes

3. Have the authors made all data underlying the findings in their manuscript fully available?

Reviewer #1: No

Reviewer #2: Yes

4. Is the manuscript presented in an intelligible fashion and written in standard English?

Reviewer #1: Yes

Reviewer #2: Yes

5. Review Comments to the Author

Reviewer #1: The aim of the manuscript was to assess the dimensionality of the "Formal Thought Disorder Self- Report Scale. Although the topic of the manuscript seems to be of scientific interest, here are some comments which could improve the quality of the manuscript.

- I kindly ask the authors to add to the description of the FTD-SS what lower and higher score reflect (e.g., greater difficulties with communication?).

- Something with the samples is not fully convincing. The authors state that “The FTD-SS was presented as part of a larger online survey.”

So, it seems legit to ask whether these samples were in reality one big sample which was splitted into three? If this is the case, I kindly ask the authors to repeat the analyses on the whole sample and compare the different models fit using the nested model approach, because I cannot see the point of splitting the sample. If this is not the case, then I would like to see whether there are any statistical significant differences in socio-demographics among these samples.

- Line 193: Add the values of kurtosis and skewness

- Line 222: I did not understand whether these 9 items (the 7 items which did not load significantly on any of the three factors and the 2 items which did not load on the common factor), where still included in the scale and further analyses? If yes, please explain which item were kept (just the 2 items, or the 7 items, or all the 9 items), and why this decision was made.

- Moreover, it is not clear whether the three factors identified in your model, reflect the structure of the factors reported by Barrera? In other words, the items loading on your factors, also load together in the model identified by Barrera? If not, please what could be a possible explanation and why did you change the name of the factors? Finally, please adapt the discussion on the basis of these results.

- Please move “Study 2” before the aim of the study because the readers cannot tell where the discussion of the previous study end and where the aim of the new study begin.

- Line 322: I do not fully understand the rationale behind testing a second-order model on the bifactor model. Could the authors explain this decision?

- Line 344: The authors state “Notably, however, none of the participants in Sample 2 endorsed the fourth response category for item 15. The distribution for the item was similar in Sample 3, with only three participants endorsing the fourth response category. Thus, for comparability, the third and fourth response categories for this item were collapsed together in Sample 3.”. However, do the authors checked if reducing the number of categories for an item could affect model fit? Secondly, was this procedure also applied in sample 2? If not, why? Thirdly, if this procedure was in fact applied in Sample 2, it was not in sample 1. Therefore, the first aim stated by the authors for this second study “The first aim was to determine whether the bifactor model that was found in Study 1 could adequately represent responses on the FTD-SS obtained in a second student sample, and to explore whether the fit of this model surpassed that of an alternative second-order model, as well as the correlated three-factor solution reported by Barrera et al., (60) and a simple unidimensional model.”, may not be demonstrated.

Lastly, since this procedure was not performed on the sample 1, the second aim indicated by the authors “The second aim was to explore the generalizability of the best fitting model by investigating the invariance of the confirmed factor solution in a third sample derived using a different recruitment strategy.” may not be demonstrated. The fit of a model with a reduced number of categories for an item cannot be compared to the fit of a model with no reduced item categories.

I suggest the authors to not reduce the number of categories and repeat the analyses.

- Line 324: The authors state “The best fitting of these models was then applied to the responses collected in Sample 3 to demonstrate form invariance across the two samples.”. However, it is not clear to which samples they are referring to? sample 1 and 2? or 2 and 3?

Moreover, they reported testing for “form invariance”. Do they perhaps mean configural invariance? I suggest the authors to explain that part better.

Anyway, I agree with the authors that is important to test the invariance of the model, however, I would suggest to compare males and females or to re-organize the analyses conducted on each sample (so maybe they could use two sample to test fo invariance), or to collect new data.

- Move “Study 3” (see comment for “Study 2”).

- Moreover, since the aim of study 3 was to assess the validity of FTD-SS with other measure of psychopathology, it is not clear what is the rationale (or aim) behind the regression analyses conducted. I kindly ask the author to explain the need for regression analyses, otherwise I suggest the authors to drop these analyses. There are already multiple analyses in the manuscript, therefore adding analyses not supported by a clear aim could make the manuscript even harder to read.

-Line 409: It is not clear what do the authors mean with “A combination of free-response and multiple-choice questions were presented to record the following demographic information…”. I kindly ask the authors to explain better and rephrase.

- Please add for each measure administered: (1) the scale (e.g., Likert scale from …); (2) what higher scores mean; (3) indices of internal consistency (e.g., Cronbach’s alpha or ordinal alpha).

- Since the authors reported that they reduced the number of categories for item 15 in for sample 2 and 3 in the second study, but not for sample 1 in the first study, and, moreover, since for study 3 they used pooled data from all the samples, it is not clear if they reduced the number of categories also for sample 1. However, I suggest the author not to reduce the number of categories (see previous comment) and repeat the analyses for study 3.

General consideration: overall the manuscript is hard to read, mainly because its organization and the high number of statistical analyses. I suggest the author to re-organize the structure of the manuscript, by both re-organize the sections of the different studies, and computing the analyses on the whole sample. If the last suggestion is not possible, then I recommend the authors to re-organize the analyses conducted on each sample, without being too redundant.

Reviewer #2: Thank you. This is a solid report, part of a consistent line of research of a scientific and clinical importance. The statistical and psychometric analyses are complex, sound, and well presented. The implications and potential developments of the findings are well discussed. Just a minor detail: there may be a typo in line 633.

6. PLOS authors have the option to publish the peer review history of their article (what does this mean?). If published, this will include your full peer review and any attached files.

Reviewer #1: No

Reviewer #2: No

---

## [Author Response · Author response to Decision Letter 0]

16 Jan 2022

Reviewer #1: 

The aim of the manuscript was to assess the dimensionality of the "Formal Thought Disorder Self- Report Scale. Although the topic of the manuscript seems to be of scientific interest, here are some comments which could improve the quality of the manuscript.

(Response)

Thank you for your feedback!

- I kindly ask the authors to add to the description of the FTD-SS what lower and higher score reflect (e.g., greater difficulties with communication?).

(Response)

We have added this into the description of the FTD-SS on page 8 (lines 158-160), indicating that higher scores reflect more frequent difficulties with communication.

- Something with the samples is not fully convincing. The authors state that “The FTD-SS was presented as part of a larger online survey.”

So, it seems legit to ask whether these samples were in reality one big sample which was splitted into three? If this is the case, I kindly ask the authors to repeat the analyses on the whole sample and compare the different models fit using the nested model approach, because I cannot see the point of splitting the sample. If this is not the case, then I would like to see whether there are any statistical significant differences in socio-demographics among these samples.

(Response)

As mentioned in the introduction (pages 6-7, lines 120-127), only one study had previously investigated the dimensionality of non-clinical disorganised or constrained thought and speech (DCTS), and this study used a Principal Components Analysis (Barrera et al., 2015). Moreover, the dimensionality of clinical DCTS is not yet agreed upon. Thus, because there was no theoretical basis to guide a CFA, we initially set out to investigate the dimensionality of the FTD-SS using exploratory (i.e. data-driven) analyses based on polychoric correlations (as appropriate for ordinal data). 

Since the bifactor model from the EFA of Study 1 diverged somewhat from Barrera et al.’s (2015) correlated three-factors model, we decided to continue data collection so that we could use CFA to assess the replicability of the model in a new sample, and so that we could determine the improvement in fit over competing models. As such, the aim of Study 2 was motivated directly from the findings of Study 1, and it would not be appropriate to combine the samples/analysis between Study 1 and Study 2.

This approach has been advocated as good model development practice (Byrne, 2010). In particular, EFA should be used when the links between observed and latent variables are uncertain, followed by CFA to validate the measurement model suggested by the EFA (pg. 5, Byrne, 2010). Using separate samples for the EFA and CFA ensures that there is a robust validation process.

On the other hand, the recruitment of the third sample was done in an effort to improve the generalisability of the current findings by being less reliant upon student participants. Although we managed to recruit more non-students and more males through Prolific, the sampled FTD-SS scores (and levels of schizotypy more generally) were highly similar. Therefore, in-light of your feedback, we have re-conducted the CFA combining the second and third samples (now referred to as “Sample 2”). The limits to generalisability are still mentioned in the discussion. 

- Line 193: Add the values of kurtosis and skewness

(Response)

These values for each item have been added to Table 2, and this addition has been referred to in-text (page 15, lines 198-199).

- Line 222: I did not understand whether these 9 items (the 7 items which did not load significantly on any of the three factors and the 2 items which did not load on the common factor), where still included in the scale and further analyses? If yes, please explain which item were kept (just the 2 items, or the 7 items, or all the 9 items), and why this decision was made.

(Response)

All 9 items were retained in the subsequent CFA in Study 2. There were several reasons for this. Firstly, as mentioned in the materials section of Study 1 (page 8, lines 152-156), all 29 items from the FTD-SS were specifically designed based on distinct clinical symptoms of thought disorder, and so are important conceptually. Moreover, the items were retained from an initial pool of 52 items based on evidence of their reliability and level of endorsement in a clinical sample of people with schizophrenia (Barrera et al., 2008). These items, including the 9 in question here, were found to load significantly in the factor solution reported for this clinical sample, as well as in the correlated three-factors model reported by Barrera et al. (2015) in their non-clinical sample. It is also conceptually important to replicate the factor loadings for these 9 items because non-significant loadings might mean that these indicators are less important in non-clinical samples than in people with schizophrenia. Hence, we were reluctant to remove items without first confirming these loadings in the second sample.

Moreover, one of the aims of the CFA in Study 2 was to compare the bifactor model with other models, including Barrera et al.’s (2015) model, and this requires the retention of all 29 items, with weak/non-significant loadings attenuating support for a model. The 2 items that did not load significantly on the general factor might be considered more problematic for the bifactor model because of the importance/dominance of the general factor (based on the closeness-to-unidimensionality indices from the EFA provided in the supporting information section). Nonetheless, in the CFA, we found that all 29 items (including items 7 and 14) loaded significantly on the bifactor model, and we confirmed that dominance of the general factor. Thus, these items were included in the total scores for the regression analyses of Study 3.

Ultimately, there might be an argument for the removal of items from the FTD-SS as, in addition to the factor loadings, the large internal consistency coefficients in the current samples could indicate redundancy. However, the removal of particular items here would be premature because they depend on the theoretical interpretation of the bifactor model. For example, if the group factors represent nuisance variables unrelated to DCTS, then the 7 items that do not load significantly on any of the group factors might be considered better indicators of DCTS than the remaining items. However, if the group factors represent theoretically meaningful dimensions of DCTS, then these items are also important and might be considered better indicators of their group factors. These possibilities illustrate the importance of determining the convergent validity of the factors, which we began to do in Study 3. Nevertheless, further research is required, as we have indicated in the general discussion (page 40-41).

- Moreover, it is not clear whether the three factors identified in your model, reflect the structure of the factors reported by Barrera? In other words, the items loading on your factors, also load together in the model identified by Barrera? If not, please what could be a possible explanation and why did you change the name of the factors? Finally, please adapt the discussion on the basis of these results.

(Response)

Twenty-two items showed significant group factor loadings within the bifactor model. With one exception (item 4), these items showed significant loadings on the same factors as those reported by Barrera et al. (2015). As alluded to in the discussion (page 19, line 263-265), the main difference was that the introduction of the general factor resulted in 7 items that no longer loaded significantly on any group factor. Notably, these 7 items tended to be amongst those with the lowest factor loadings in Barrera et al.’s (2015) correlated three-factors model. 

Despite these similarities, we elected to change the factor labels from those of Barrera et al. (2015) in favour of more descriptive terms, in-line with our previous work (Sumner et al., 2020). Barrera et al. (2015) themselves were inconsistent in their labelling of factor 2, which they referred to both as “Alogia” and “Conversational Ability”. Moreover, their use of “Working Memory Deficit” implies that the factor results from issues with working memory function, which has not yet been established. We have now explained this re-labelling in the results section of Study 1 (page 16, lines 233-236).

- Please move “Study 2” before the aim of the study because the readers cannot tell where the discussion of the previous study end and where the aim of the new study begin.

(Response)

We have moved the title, as suggested.

- Line 322: I do not fully understand the rationale behind testing a second-order model on the bifactor model. Could the authors explain this decision?

(Response)

Indeed, the two models appear similar, both representing three group factors and a general factor. However, the two models differ in the specified relationships between the group and general factors, and so yield different conceptual representations of multidimensionality (see Reise et al., 2010). In the bifactor model, direct relationships are modelled between the observed item variance and the general factor, and the group factors explain additional item variance that is not accounted for by the general factor. One intuitive interpretation of the group factors is that they represent nuisance variables that interfere with the measurement of the general factor (Reise et al., 2010). By contrast, in the second-order hierarchical model, the general factor is not directly related to the item variances. Instead, the general factor accounts for the variance that is common to the group factors. 

We attempted to explain this distinction in the discussion for Study 1 using intelligence models as an example. The explanation has been re-worded slightly in an effort to clarify the rationale (page 21, lines 301-305, 310-312). The differences between the models are also depicted in Figures 3 and 4.

- Line 344: The authors state “Notably, however, none of the participants in Sample 2 endorsed the fourth response category for item 15. The distribution for the item was similar in Sample 3, with only three participants endorsing the fourth response category. Thus, for comparability, the third and fourth response categories for this item were collapsed together in Sample 3.”. However, do the authors checked if reducing the number of categories for an item could affect model fit? Secondly, was this procedure also applied in sample 2? If not, why? Thirdly, if this procedure was in fact applied in Sample 2, it was not in sample 1. Therefore, the first aim stated by the authors for this second study “The first aim was to determine whether the bifactor model that was found in Study 1 could adequately represent responses on the FTD-SS obtained in a second student sample, and to explore whether the fit of this model surpassed that of an alternative second-order model, as well as the correlated three-factor solution reported by Barrera et al., (60) and a simple unidimensional model.”, may not be demonstrated.

Lastly, since this procedure was not performed on the sample 1, the second aim indicated by the authors “The second aim was to explore the generalizability of the best fitting model by investigating the invariance of the confirmed factor solution in a third sample derived using a different recruitment strategy.” may not be demonstrated. The fit of a model with a reduced number of categories for an item cannot be compared to the fit of a model with no reduced item categories.

I suggest the authors to not reduce the number of categories and repeat the analyses.

- Line 324: The authors state “The best fitting of these models was then applied to the responses collected in Sample 3 to demonstrate form invariance across the two samples.”. However, it is not clear to which samples they are referring to? sample 1 and 2? or 2 and 3?

Moreover, they reported testing for “form invariance”. Do they perhaps mean configural invariance? I suggest the authors to explain that part better.

Anyway, I agree with the authors that is important to test the invariance of the model, however, I would suggest to compare males and females or to re-organize the analyses conducted on each sample (so maybe they could use two sample to test fo invariance), or to collect new data.

(Response)

We thank the reviewer for this feedback. Combining the samples in Study 2 has removed these problems and brought the analysis closer to its aim. The CFA has been re-conducted with all response categories for item 15 intact. 

Unfortunately, we are not able to test configural invariance in this current data (Sample 2) because of insufficient data (i.e. unequal group sizes) across our demographics. We have noted this in the discussion as an important avenue for future research (page 42, lines 680-682).

- Move “Study 3” (see comment for “Study 2”).

(Response)

We have moved the title, as suggested.

- Moreover, since the aim of study 3 was to assess the validity of FTD-SS with other measure of psychopathology, it is not clear what is the rationale (or aim) behind the regression analyses conducted. I kindly ask the author to explain the need for regression analyses, otherwise I suggest the authors to drop these analyses. There are already multiple analyses in the manuscript, therefore adding analyses not supported by a clear aim could make the manuscript even harder to read.

(Response)

The underlying theoretical motivation behind the current study was that the FTD-SS might be a more sensitive measure of DCTS in schizotypy, when compared to other schizotypy measures that are typically used, because the FTD-SS might better capture the multidimensionality of DCTS that has been reported in clinical samples (pages 4-5, lines 74-86; page 7, lines 128-141). Thus, the aim of the study was to investigate the construct validity of the FTD-SS, with investigations of the dimensionality being a main focus (for Study 1 and Study 2). The regression analyses in Study 3 are, nevertheless, integral to the main theoretical motivation because they help evaluate how the FTD-SS performs in comparison to other measures of schizotypy. Notably, very few studies have investigated the convergent validity of self-report measures of DCTS, and our previous study is the only one to do so for the FTD-SS (Sumner et al., 2020). Since the schizotypy measures explained around half the variance in FTD-SS scores in the current dataset, and this convergence was not accounted for by neuroticism or extraversion (which have been suggested to contaminate self-report measures of DCTS), the findings of Study 3 should foster additional research into the potential incremental validity of the FTD-SS. 

-Line 409: It is not clear what do the authors mean with “A combination of free-response and multiple-choice questions were presented to record the following demographic information…”. I kindly ask the authors to explain better and rephrase.

(Response)

The wording has been removed to avoid confusion (page 29, lines 445-446). 

- Please add for each measure administered: (1) the scale (e.g., Likert scale from …); (2) what higher scores mean; (3) indices of internal consistency (e.g., Cronbach’s alpha or ordinal alpha).

(Response)

We had already calculated coefficient omegas to test internal consistency in Study 2, as recommended by several authors under most conditions (e.g. Reise et al., 2010; Zinbarg et al., 2005). However, for comparability, we have added total ordinal alpha coefficients for the FTD-SS in both Sample 1 (page 19, lines 260-261) and Sample 2 (page 27, lines 389-390). Ordinal coefficient alphas for the other scales were provided in the supporting information section, and these have now been moved into the manuscript (pages 30-32). 

The response scales and interpretation of scale scores have also been added for all of the measures (page 8, lines 158-160; pages 30-32).

- Since the authors reported that they reduced the number of categories for item 15 in for sample 2 and 3 in the second study, but not for sample 1 in the first study, and, moreover, since for study 3 they used pooled data from all the samples, it is not clear if they reduced the number of categories also for sample 1. However, I suggest the author not to reduce the number of categories (see previous comment) and repeat the analyses for study 3.

(Response)

We have combined the two samples in Study 2, removing the need to collapse the number of categories for item 15. Thus, all analyses are now consistent across all samples, including the analyses conducted in Study 3.

General consideration: overall the manuscript is hard to read, mainly because its organization and the high number of statistical analyses. I suggest the author to re-organize the structure of the manuscript, by both re-organize the sections of the different studies, and computing the analyses on the whole sample. If the last suggestion is not possible, then I recommend the authors to re-organize the analyses conducted on each sample, without being too redundant.

(Response)

We again thank you for your constructive feedback and we hope that the clarifications made are sufficient to improve the readability of the analyses. Unfortunately, we cannot re-structure the analyses because, as mentioned, the CFA of Study 2 is based upon the findings from the EFA of Study 1, and because the analysis in Study 3 are based on the findings from Study 1 and Study 2 (i.e. continued support for the bifactor model). However, we have combined the sample for Study 2 and re-conducted the CFA, based on your feedback. Hopefully, with the edits made, you find the analyses easier to follow.

Reviewer #2: 

Thank you. This is a solid report, part of a consistent line of research of a scientific and clinical importance. The statistical and psychometric analyses are complex, sound, and well presented. The implications and potential developments of the findings are well discussed. Just a minor detail: there may be a typo in line 633.

(Response)

We thank you for your positive feedback, and for identifying the typo. This has been corrected.

Additional References:

(References used here but not in the manuscript)

Byrne (2010). Structural Equation Modelling with AMOS. Basic concepts, applications and programming. Second edition. New York, Routledge.

Zinbarg, R. E., Revelle, W., Yovel, I., & Li, W. (2005). Cronbach’s α, Revelle’s β, and McDonald’s ωH: Their relations with each other and two alternative conceptualizations of reliability. Psychometrika, 70(1), 123-133.

---

## [Decision Letter · Decision Letter 1]

14 Mar 2022

PONE-D-21-24412R1Assessing the dimensionality of scores derived from the Formal Thought Disorder Self-Report Scale in schizotypyPLOS ONE

Dear Dr. Sumner,

Thank you for submitting your manuscript to PLOS ONE. After careful consideration, we feel that it has merit but does not fully meet PLOS ONE’s publication criteria as it currently stands. Therefore, we invite you to submit a revised version of the manuscript that addresses the points raised during the review process.

Apr 28 2022 11:59PM. If you will need more time than this to complete your revisions, please reply to this message or contact the journal office at plosone@plos.org. Please include the following items when submitting your revised manuscript:A rebuttal letter that responds to each point raised by the academic editor and reviewer(s). You should upload this letter as a separate file labeled 'Response to Reviewers'.A marked-up copy of your manuscript that highlights changes made to the original version. You should upload this as a separate file labeled 'Revised Manuscript with Track Changes'.An unmarked version of your revised paper without tracked changes. You should upload this as a separate file labeled 'Manuscript'.

We look forward to receiving your revised manuscript.

Kind regards,

Marco Innamorati

Academic Editor

PLOS ONE

Reviewers' comments:

Reviewer's Responses to Questions

**Comments to the Author**

1. If the authors have adequately addressed your comments raised in a previous round of review and you feel that this manuscript is now acceptable for publication, you may indicate that here to bypass the “Comments to the Author” section, enter your conflict of interest statement in the “Confidential to Editor” section, and submit your "Accept" recommendation.

Reviewer #1: (No Response)

Reviewer #2: All comments have been addressed

2. Is the manuscript technically sound, and do the data support the conclusions?

Reviewer #1: No

Reviewer #2: (No Response)

3. Has the statistical analysis been performed appropriately and rigorously? 

Reviewer #1: No

Reviewer #2: (No Response)

4. Have the authors made all data underlying the findings in their manuscript fully available?

Reviewer #1: No

Reviewer #2: (No Response)

5. Is the manuscript presented in an intelligible fashion and written in standard English?

Reviewer #1: Yes

Reviewer #2: (No Response)

6. Review Comments to the Author

Reviewer #1: STUDY 1

- The authors state that they combined the second and third sample, which now is referred as “Sample 2”. I support this decision because there is no need to collect two more samples in order to assess replicability and generalizability of the model.

However, during the first review process I also suggested to the authors to assess whether there are socio-demographics differences between the samples. I did not find any tables or results showing whether the two samples are statistically different or not. So, again, I kindly ask the authors to conduct these analyses. If the two samples are statistically significant, I suggest the authors to repeat the EFA on both samples to assess whether these differences had affected the models whatsoever.

In any case, if there are statistical differences for socio-demographics, I suggest the authors to discuss the results in line of these differences.

- I am not totally satisfied with the explanation given by the authors on the decision to keep the nine items in the test. If seven items did not load on any of the group factors, I suggest the authors to perform a deeper item analysis (e.g. Mokken analysis) and eventually remove these problematic items. The presence of seven problematic items is too big to ignore and decide that these items should still be included in the scale without a proper item analysis.

The same goes for the two items that did not load on the general factor.

- The authors state “Because the factors found in the initial analysis appeared to be highly correlated with one another (≥ 0.56), a follow-up exploratory linear bifactor model was conducted…”.

However, this is not in line with what a bifactor model aims at assessing. A bifactor model aims at assessing unidimensionality: when the items report a greater load on the general common factor than on the other group factors, than it can be assumed that the construct is essentially unidimensional, and this is not the case for the bifactor model reported in the manuscript. Some items show a greater loading on the general common factor, while others show a greater factor loading on the other factors extracted.

Moreover, Reise (2012) reported “Exploratory analyses allow researchers to identify potential modeling problems directly rather than indirectly through post hoc inspection of fit and modification indices after estimating a confirmatory model.” (Reise, S. P. (2012). Invited paper: the rediscovery of bifactor measurement models. Multivariate Behav. Res. 47, 667–696. doi: 10.1080/00273171.2012.715555). Which means that a bifactor model cannot be the final solution. For these reasons, I suggest the authors to use the bifactor model to identify the presence of problematic items (see previous comment), and to modify the model if needed (since two items did not basically load on the general common factor (0.15 and 0.24)).

Moreover, the author state “The addition of a fourth factor in the bifactor model explained an extra 4.72% of the variance ”. That is obviously true because the more factors there are, the more variance is explained, and that is why we have to be careful with factor analysis in trying to find a balance between the number of factors and the variance explained. Including a general common factor, in addition to the three group factors already extracted, is not a valid theoretical solution. A general common factor cannot be considered, for example, as a second-order factor, which can be used for theoretical explanation.

Lastly, the authors state that in the CFA all items significantly loaded on the bifactor model. However, if the EFA did not indicate a good fit to the data, then it is pointless to perform a CFA (since the aim of the CFA is to confirm the results found by the EFA). So, I kindly suggest the authors to reconduct the EFA after the analyses on the problematic items.

- Because of the previous comment, I kindly suggest the authors to remove the bifactor model fit from Table 3. Table 3 should be a recap of all the possible model solutions (since it was established that a bifactor model cannot be considered a model solution). For model comparison the authors could report the AIC and BIC values and choose the most suitable model on the basis of their values.

STUDY 2

- “Line 322: I do not fully understand the rationale behind testing a second-order model on the bifactor model. Could the authors explain this decision?” Response: “Indeed, the two models appear similar, both representing three group factors and a general factor. However, the two models differ in the specified relationships between the group and general factors, and so yield different conceptual representations of multidimensionality (see Reise et al., 2010). In the bifactor model, direct relationships are modelled between the observed item variance and the general factor, and the group factors explain additional item variance that is not accounted for by the general factor. One intuitive interpretation of the group factors is that they represent nuisance variables that interfere with the measurement of the general factor (Reise et al., 2010). By contrast, in the second-order hierarchical model, the general factor is not directly related to the item variances. Instead, the general factor accounts for the variance that is common to the group factors.

We attempted to explain this distinction in the discussion for Study 1 using intelligence models as an example. The explanation has been re-worded slightly in an effort to clarify the rationale (page 21, lines 301-305, 310-312). The differences between the models are also depicted in Figures 3 and 4. ”

I know what a second-order hierarchical model is, however, that is not what I asked when I made the comment on the first review process. In the original version of the manuscript it was reported that new models were tested on sample 2, among which “… and a second-order model based on this bifactor model”(former line 322). Maybe it was the wording that was misunderstanding or the fact that the manuscript was (and still is) hard to read, but it seemed like that a second-order bifactor model was being tested, and that is pointless to test.

- The authors replied to my comment saying that they were unable to test configural invariance because of insufficient data, so I wanted to know what kind of invariance they tested? What is “form invariance”? Please, could the authors provide some information?

General consideration: I kindly ask the authors to re-organize the manuscript in light of the comments made about the bifactor model. Moreover, I kindly suggest the authors to take into consideration the fact of rearranging the manuscript order on the basis of the results of the analyses requested in the first 2 comments (the analyses on the differences in socio-demographic variables between the two samples was already requested during the first review process). Hence, one possible solution could be to combine Study 1 and Study 2 (since the main analysis conducted is factor analysis), which will become “Study 1” and then Study 3, which will become “Study 2”. If the authors do not agree with this suggestion, I kindly ask them to add the new analyses in the most suitable place in the manuscript so it will not make it harder to read.

Reviewer #2: (No Response)

7. PLOS authors have the option to publish the peer review history of their article (what does this mean?). If published, this will include your full peer review and any attached files.

Reviewer #1: No

Reviewer #2: No

---

## [Author Response · Author response to Decision Letter 1]

21 Jul 2022

We thank Reviewer 1 for taking the time to go through our manuscript a second time, and offering their valuable feedback. We have tried our best to implement all suggestions in some way.

- The authors state that they combined the second and third sample, which now is referred as “Sample 2”. I support this decision because there is no need to collect two more samples in order to assess replicability and generalizability of the model.

However, during the first review process I also suggested to the authors to assess whether there are socio-demographics differences between the samples. I did not find any tables or results showing whether the two samples are statistically different or not. So, again, I kindly ask the authors to conduct these analyses. If the two samples are statistically significant, I suggest the authors to repeat the EFA on both samples to assess whether these differences had affected the models whatsoever.

In any case, if there are statistical differences for socio-demographics, I suggest the authors to discuss the results in line of these differences.

Between-group differences have now been tested and are presented in Table 1. We have also discussed the potential sample differences as potentially influencing the fit of the unidimensional model in the CFA with Sample 2, and so this is a study limitation (lines 538-546, page 30). Although we have not reported this, the EFA results in both samples were somewhat similar. Although the parallel analysis indicated 2 factors in Sample 2, Heywood cases were still evident in the multidimensional solutions. There were also similar signs of item redundancy in both samples, and the case for unidimensionality strengthened with the removal of these items in both samples (mentioned lines 360-375, pages 20 and 21). 

Note too that we now discuss demographic influences and sampling characteristics extensively when comparing our findings with those of Barrera et al. (2015; see lines 425-478, pages 26 and 27).

- I am not totally satisfied with the explanation given by the authors on the decision to keep the nine items in the test. If seven items did not load on any of the group factors, I suggest the authors to perform a deeper item analysis (e.g. Mokken analysis) and eventually remove these problematic items. The presence of seven problematic items is too big to ignore and decide that these items should still be included in the scale without a proper item analysis.

The same goes for the two items that did not load on the general factor.

We agree that an item analysis is required but maintain that it is still important to show the performance of the original scale with all 29 items. As mentioned above, we discuss the differences in the item responses sampled in the current study (both Sample 1 and Sample 2) compared to those sampled by Barrera et al. (2015), despite seemingly similar recruitment avenues and sample demographics between the two studies. Reasons for these differences are not yet clear. However, it makes sense to present this evidence and the inability to replicate their correlated three factors model before revising the scale, since the measure has already been created and used. 

Nevertheless, there were some signs of item redundancy in both samples. We have now reported the largest inter-item polychoric correlations that we found in Sample 1, and note that these items also produced large residual correlations across all dimension-reduction techniques in both samples. We then indicated the effect of removing these items on the evidence in support of unidimensionality (see lines 354-375, pages 20 and 21). This promotes the conclusion that revisions to the scale are necessary, at least in samples with levels of endorsement comparable to those in the current study.

Moreover, based on your feedback below, we have revised the description of the results from the bifactor model (and the added Rasch analysis) to indicate that we are evaluating the evidence in support of the plausibility of subscales (i.e. multidimensionality) and the evidence in support of the plausibility of the overall score (i.e. unidimensionality; following the approach outlined by Dunn et al., 2020). In this context, the high number of items that did not load significantly on any group factor was taken as support for unidimensionality. On the other hand, the two items with non-significant loadings on the general factor, and the poor fit of the unidimensional model in Sample 2, both suggest the need for item revisions.

We have elaborated our point in the discussion that some of the FTD-SS items do not seem to be appropriate for non-clinical samples, at least in our non-clinical samples, given that some items exhibited low levels of endorsement (high positive skew). We then suggest that a deeper item analysis be conducted in future studies. We hope to conduct these analyses as a separate manuscript, using the Rasch analysis more extensively (beyond testing the assumption of unidimensionality).

- The authors state “Because the factors found in the initial analysis appeared to be highly correlated with one another (≥ 0.56), a follow-up exploratory linear bifactor model was conducted…”.

However, this is not in line with what a bifactor model aims at assessing. A bifactor model aims at assessing unidimensionality: when the items report a greater load on the general common factor than on the other group factors, than it can be assumed that the construct is essentially unidimensional, and this is not the case for the bifactor model reported in the manuscript. Some items show a greater loading on the general common factor, while others show a greater factor loading on the other factors extracted.

Moreover, Reise (2012) reported “Exploratory analyses allow researchers to identify potential modeling problems directly rather than indirectly through post hoc inspection of fit and modification indices after estimating a confirmatory model.” (Reise, S. P. (2012). Invited paper: the rediscovery of bifactor measurement models. Multivariate Behav. Res. 47, 667–696. doi: 10.1080/00273171.2012.715555). Which means that a bifactor model cannot be the final solution. For these reasons, I suggest the authors to use the bifactor model to identify the presence of problematic items (see previous comment), and to modify the model if needed (since two items did not basically load on the general common factor (0.15 and 0.24)).

Moreover, the author state “The addition of a fourth factor in the bifactor model explained an extra 4.72% of the variance ”. That is obviously true because the more factors there are, the more variance is explained, and that is why we have to be careful with factor analysis in trying to find a balance between the number of factors and the variance explained. Including a general common factor, in addition to the three group factors already extracted, is not a valid theoretical solution. A general common factor cannot be considered, for example, as a second-order factor, which can be used for theoretical explanation.

Lastly, the authors state that in the CFA all items significantly loaded on the bifactor model. However, if the EFA did not indicate a good fit to the data, then it is pointless to perform a CFA (since the aim of the CFA is to confirm the results found by the EFA). So, I kindly suggest the authors to reconduct the EFA after the analyses on the problematic items.

We thank the reviewer for this feedback. The description of our findings has been revised to better indicate evidence for and against unidimensionality, and removed mistaken statements suggesting that the bifactor model was a final solution. We had already concluded that the best evidence from the bifactor model was of essential unidimensionality (a conclusion that was confirmed by the corresponding author for the FACTOR program when I wrote to him to clarify the output), suggesting that a total/overall score would be the most meaningful measure derived from the questionnaire. We had initially based this on the closeness-to-unidimensionality indices that are presented as supporting information. We have now bolstered this conclusion by further considering the pattern of factor loadings in the bifactor model, conducting a Rasch analysis, and assessing the influence of item redundancy. We have also scaled back the CFA to simply show that the unidimensional model was a poor fit in a second sample. Perhaps this is unsurprising, though it serves to reproduce the lower levels of item endorsement that contrast the data presented by Barrera et al. (2015). It also affirms the need for a more in-depth analysis of the items and revisions of the scale.

- Because of the previous comment, I kindly suggest the authors to remove the bifactor model fit from Table 3. Table 3 should be a recap of all the possible model solutions (since it was established that a bifactor model cannot be considered a model solution). For model comparison the authors could report the AIC and BIC values and choose the most suitable model on the basis of their values.

We have removed Table 3. As you mentioned above, it makes little sense to test models that were not supported by the EFA. Since the balance of evidence from the exploratory analyses favoured a unidimensional model, this is the only model now tested using the CFA. Although the figures depicting the unidimensional model, Barrera et al.’s correlated three-factor model and the bifactor model have been retained, these are for illustrative purposes.

STUDY 2

- “Line 322: I do not fully understand the rationale behind testing a second-order model on the bifactor model. Could the authors explain this decision?” Response: “Indeed, the two models appear similar, both representing three group factors and a general factor. However, the two models differ in the specified relationships between the group and general factors, and so yield different conceptual representations of multidimensionality (see Reise et al., 2010). In the bifactor model, direct relationships are modelled between the observed item variance and the general factor, and the group factors explain additional item variance that is not accounted for by the general factor. One intuitive interpretation of the group factors is that they represent nuisance variables that interfere with the measurement of the general factor (Reise et al., 2010). By contrast, in the second-order hierarchical model, the general factor is not directly related to the item variances. Instead, the general factor accounts for the variance that is common to the group factors.

We attempted to explain this distinction in the discussion for Study 1 using intelligence models as an example. The explanation has been re-worded slightly in an effort to clarify the rationale (page 21, lines 301-305, 310-312). The differences between the models are also depicted in Figures 3 and 4. ”

I know what a second-order hierarchical model is, however, that is not what I asked when I made the comment on the first review process. In the original version of the manuscript it was reported that new models were tested on sample 2, among which “… and a second-order model based on this bifactor model”(former line 322). Maybe it was the wording that was misunderstanding or the fact that the manuscript was (and still is) hard to read, but it seemed like that a second-order bifactor model was being tested, and that is pointless to test.

These sections have been removed. 

- The authors replied to my comment saying that they were unable to test configural invariance because of insufficient data, so I wanted to know what kind of invariance they tested? What is “form invariance”? Please, could the authors provide some information?

“Form invariance” was used erroneously. We had hoped to explore whether the factor solutions differed across potentially important demographic variables (e.g. gender). Originally, the second sample was collected using REP and the third sample was collected using Prolific (these groups are now combined into Sample 2). Prolific was used for recruitment in the hope of collecting a more representative non-clinical sample, including a greater proportion of males and less students. The first step is to test configural invariance, before successively constraining model parameters to be equal to test stricter forms of invariance. However, our sample sizes were not large enough to test invariance between levels of any the demographic variables assessed. 

General consideration: I kindly ask the authors to re-organize the manuscript in light of the comments made about the bifactor model. Moreover, I kindly suggest the authors to take into consideration the fact of rearranging the manuscript order on the basis of the results of the analyses requested in the first 2 comments (the analyses on the differences in socio-demographic variables between the two samples was already requested during the first review process). Hence, one possible solution could be to combine Study 1 and Study 2 (since the main analysis conducted is factor analysis), which will become “Study 1” and then Study 3, which will become “Study 2”. If the authors do not agree with this suggestion, I kindly ask them to add the new analyses in the most suitable place in the manuscript so it will not make it harder to read.

We have now re-structured the manuscript, removing the Study 1/Study 2/Study 3 format and organising the results by “exploratory dimension-reduction analyses”, “confirmatory factor analysis”, “potential item redundancy”, and “convergent relationships with schizotypy and the influence of demographics”. The between-group comparisons have been provided in Table 1.

Reviewer #2: (No Response)

---

## [Decision Letter · Decision Letter 2]

29 Aug 2022

PONE-D-21-24412R2Assessing the dimensionality of scores derived from the Formal Thought Disorder Self-Report Scale in schizotypyPLOS ONE

Dear Dr. Sumner,

Thank you for submitting your manuscript to PLOS ONE. After careful consideration, we feel that it has merit but does not fully meet PLOS ONE’s publication criteria as it currently stands. Therefore, we invite you to submit a revised version of the manuscript that addresses the points raised during the review process.

We look forward to receiving your revised manuscript.

Kind regards,

Marco Innamorati

Academic Editor

PLOS ONE

Journal Requirements:

Reviewers' comments:

Reviewer's Responses to Questions

**Comments to the Author**

1. If the authors have adequately addressed your comments raised in a previous round of review and you feel that this manuscript is now acceptable for publication, you may indicate that here to bypass the “Comments to the Author” section, enter your conflict of interest statement in the “Confidential to Editor” section, and submit your "Accept" recommendation.

Reviewer #1: (No Response)

2. Is the manuscript technically sound, and do the data support the conclusions?

Reviewer #1: Partly

3. Has the statistical analysis been performed appropriately and rigorously? 

Reviewer #1: No

4. Have the authors made all data underlying the findings in their manuscript fully available?

Reviewer #1: Yes

5. Is the manuscript presented in an intelligible fashion and written in standard English?

Reviewer #1: Yes

6. Review Comments to the Author

Reviewer #1: I appreciate the authors responsiveness to my comments.

However, I still a have a minor concern which was not addressed in this second review process.

The authors provided support for a unidimensional factor structure for the FTD-SS with a bifactor model, however two items did not load on the general common latent factor. Moreover, the authors still decided to keep the seven items which did not load on the specific group factors. I was wondering if the authors tried to conduct the analyses without those item. Firstly, without all the nine items, then excluding the two items which did not load on the general common factor, and lastly excluding only the seven items which did not load on the specific group factors. By doing so it's possible to assess the changes in the factor structure, and, perhaps, an improvement of the model fit.

I think this is an important step in order to justify the presence of these items in the scale.

7. PLOS authors have the option to publish the peer review history of their article (what does this mean?). If published, this will include your full peer review and any attached files.

Reviewer #1: No

---

## [Author Response · Author response to Decision Letter 2]

31 Oct 2022

Reviewer #1: 

I appreciate the authors responsiveness to my comments. However, I still a have a minor concern which was not addressed in this second review process.

The authors provided support for a unidimensional factor structure for the FTD-SS with a bifactor model, however two items did not load on the general common latent factor. Moreover, the authors still decided to keep the seven items which did not load on the specific group factors. I was wondering if the authors tried to conduct the analyses without those item. Firstly, without all the nine items, then excluding the two items which did not load on the general common factor, and lastly excluding only the seven items which did not load on the specific group factors. By doing so it's possible to assess the changes in the factor structure, and, perhaps, an improvement of the model fit.

I think this is an important step in order to justify the presence of these items in the scale.

We thank the reviewer yet again for their advice and patience. They were correct. We removed the items and repeated the factor analyses as suggested. The removal of items improved the sampling adequacy of the polychoric correlation matrix, and parallel analyses continued to indicate the presence of three factors. Nevertheless, it was only when all nine items were removed that the three-factor solution essentially achieved simple structure (see the revised Table 2 and Table C, S1 Supporting Information). 

As mentioned in the previous re-submission, we were hesitant to remove items from the scale. The FTD-SS items were originally created by Barrera et al. (2008) based on previous descriptions of symptoms of formal thought disorder in the literature. A preliminary pool of 52 items were then reduced to the final 29 items based on an ‘item analysis’ of responses obtained from a sample of people diagnosed with chronic schizophrenia, although the authors admittedly did not describe this item analysis in much detail. All 29 items had obviously also survived in Barrera et al.’s (2015) principal components analysis of responses in their non-clinical.

With respect to this hesitancy, we brought back Barrera et al.’s (2015) model with all 29 items for the confirmatory factor analysis in Sample 2. We were then able to demonstrate the fit of the three-factor model with 20 items compared to Barrera et al.’s model with 29 items, as well as unidimensional models with both 20 and 29 items (see revised Table 3, p. 20). We are, therefore, satisfied that the removal of these items was justified.

We also re-visited our removal of items due to potential item redundancy. The inter-item correlations alone in Sample 1 were not necessarily high enough to justify removing the items straight away. However, in the last re-submission, we had removed three items (1, 9 and 14) with high inter-item correlations because they also showed high residual correlations in the Rasch analysis. As might be expected, this strengthened evidence in support of unidimensionality (although the fit of the unidimensional model was still poor with these items removed in the second sample). In an effort to strengthen the evidence of item-redundancy and better justify the removal of these items, we used Ferrando et al.’s (2022) method of detecting doublets based on residuals in the exploratory factor analysis, which is now implemented in an updated version of FACTOR. Only one pair of items (items 9 and 10) were identified as potential doublets. However, the removal of one of these items alone was not sufficient to support unidimensionality. Because the justification for the removal of the three items was not a strong as the removal of the nine items based on the bifactor model, we have moved it to the supporting information section and only mentioned it only briefly in-text (see Table C, S1 Supporting Information). 

All of the exploratory factor analyses conducted, including those repeated after the removal of items, has now been presented systematically in a table within the supporting information section (see Table C, S1 Supporting Information). This includes the improvement in sampling adequacy, any changes to the results of the parallel analysis for determining the number of factors to extract, the fit indices of the resultant solution, and the presence of any cross-loadings. The loadings for the 20-item three factor model in Sample 2 derived using confirmatory factor analyses have also replaced those of the unidimensional model in the supporting information section (Table G). 

Notably, the 20-item three-factor solution produced factors that were strongly correlated with one another, and closeness-to-unidimensionality indices still supported the use of total scores (see S1 Supporting Information). Moreover, the convergent correlation analyses were almost identical (r±0.02) when FTD-SS total scores were calculated from the 20 items instead of all 29 items. Thus, minimal changes were needed to the remainder of the analyses. Convergent bivariate correlation analyses were repeated with scores from the 20-item three-factor model, and these are presented in the supporting information (Table H).

In-text, there have been several minor changes throughout the manuscript to reflect the amended factor analyses. The loadings for the exploratory bifactor model have been moved from Table 2 to the supporting information (Table A) and replaced in-text by the loadings for the final 20-item exploratory factor solution (p. 19). Figures 1, 2 and 3 have also been altered to reflect the new confirmatory factor analyses, and a fourth figure has been added (see captions p. 31). Minor changes to the abstract (p. 2; lines: 16-23), and the order and content of the discussion (p. 25-30; lines: 419-450, 475-478, 496-498, 548-552) have been made to reflect the tipping of evidence in favour of multidimensionality. The conclusions remain largely the same, however, that the summed total score remains an important score derived from the FTD-SS, and that additional item-analyses are needed in the future.

Again, we would like to thank the reviewer for their suggestion.

---

## [Decision Letter · Decision Letter 3]

8 Nov 2022

PONE-D-21-24412R3Assessing the dimensionality of scores derived from the Formal Thought Disorder Self-Report Scale in schizotypyPLOS ONE

Dear Dr. Sumner,

Thank you for submitting your manuscript to PLOS ONE. After careful consideration, we feel that it has merit but does not fully meet PLOS ONE’s publication criteria as it currently stands. Therefore, we invite you to submit a revised version of the manuscript that addresses the points raised during the review process.

We look forward to receiving your revised manuscript.

Kind regards,

Marco Innamorati

Academic Editor

PLOS ONE

Journal Requirements:

Reviewers' comments:

Reviewer's Responses to Questions

**Comments to the Author**

1. If the authors have adequately addressed your comments raised in a previous round of review and you feel that this manuscript is now acceptable for publication, you may indicate that here to bypass the “Comments to the Author” section, enter your conflict of interest statement in the “Confidential to Editor” section, and submit your "Accept" recommendation.

Reviewer #1: (No Response)

2. Is the manuscript technically sound, and do the data support the conclusions?

Reviewer #1: Yes

3. Has the statistical analysis been performed appropriately and rigorously? 

Reviewer #1: Yes

4. Have the authors made all data underlying the findings in their manuscript fully available?

Reviewer #1: Yes

5. Is the manuscript presented in an intelligible fashion and written in standard English?

Reviewer #1: Yes

6. Review Comments to the Author

Reviewer #1: I thank the authors for being responsive to my comments.

Just a minor suggestion regarding the title. Since some items have now been removed, perhaps writing "Assessing the dimensionality of scores derived from the Formal Thought Disorder Self- Report Scale - Revised (FTD-SS-R) in schizotypy" could be more appropriate. But I will leave the decision to the authors.

7. PLOS authors have the option to publish the peer review history of their article (what does this mean?). If published, this will include your full peer review and any attached files.

Reviewer #1: No

---

## [Author Response · Author response to Decision Letter 3]

11 Nov 2022

Once we had finally acquiesced to the suggestion to remove items and repeat the exploratory factor analysis, we ourselves considered whether we should subsequently refer to the scale as a revised FTD-SS. However, we decided not to for two reasons. 

Firstly, as we have stated in the discussion of the manuscript, the demographic characteristics of the two samples in the current study were comparable to those reported by Barrera et al. (2015). Yet, the average FTD-SS total scores obtained in our samples appeared to be somewhat smaller than those of Barrera et al.’s sample, from which their three correlated factors solution was found. The reasons for this are currently unclear. Therefore, we believe that future research should conduct Rasch analyses to further investigate the performance of the items of the FTD-SS in non-clinical samples, and that a revised scale should be produced in-light of this analysis and the results of the current study.

Secondly, subsequent analyses exploring the convergent validity of FTD-SS scores and the influence of demographic variables upon these scores were performed in the current study using total scores calculated from all 29 items. Repeating these analyses with scores derived from the revised 20-item scale made very little impact on the results. However, we left these analyses as they were for the reason mentioned above. Thus, it is more accurate to leave the title as it is.

---

## [Decision Letter · Decision Letter 4]

17 Nov 2022

PONE-D-21-24412R4Assessing the dimensionality of scores derived from the Formal Thought Disorder Self-Report Scale in schizotypyPLOS ONE

Dear Dr. Sumner,

Thank you for submitting your manuscript to PLOS ONE. After careful consideration, we feel that it has merit but does not fully meet PLOS ONE’s publication criteria as it currently stands. Therefore, we invite you to submit a revised version of the manuscript that addresses the points raised during the review process.

We look forward to receiving your revised manuscript.

Kind regards,

Marco Innamorati

Academic Editor

PLOS ONE

Reviewers' comments:

Reviewer's Responses to Questions

**Comments to the Author**

1. If the authors have adequately addressed your comments raised in a previous round of review and you feel that this manuscript is now acceptable for publication, you may indicate that here to bypass the “Comments to the Author” section, enter your conflict of interest statement in the “Confidential to Editor” section, and submit your "Accept" recommendation.

Reviewer #1: (No Response)

2. Is the manuscript technically sound, and do the data support the conclusions?

Reviewer #1: Partly

3. Has the statistical analysis been performed appropriately and rigorously? 

Reviewer #1: No

4. Have the authors made all data underlying the findings in their manuscript fully available?

Reviewer #1: Yes

5. Is the manuscript presented in an intelligible fashion and written in standard English?

Reviewer #1: Yes

6. Review Comments to the Author

Reviewer #1: I am sorry but I cannot address the publication of the manuscript yet.

In the previous revisions I suggested the authors to perform changes on the factor model of the FTD-SS and to consequently adapt the results of the other analysis. The authors deleted 9 nine items from the questionnaire, which improved the fit of the model, however still kept those nine item to compute total score. How can this be correct? The readers could ask why those items were used if they were not included in the model? What is the portion of the variance explained by those items? Why did the authors not want to remove those items from the total score? The decision to still keep the nine items cannot just be theoretical. If the analyses suggest that these items should be removed, then it is pointless to compute total score with those items because they do not contribute to the factor model at all. Total score needs to be computed from only the items included in the model.

If the authors wish to compute two different total scores (one with 20 items and the other with 29 items) and analyze whether there are any statistical differences, then it is ok, but they cannot use all 29 items when the model suggested the inclusion of only 20 items.

7. PLOS authors have the option to publish the peer review history of their article (what does this mean?). If published, this will include your full peer review and any attached files.

Reviewer #1: No

---

## [Author Response · Author response to Decision Letter 4]

21 Nov 2022

Note. All page and line numbers are based on tracked changes using “Simple Markup”.

Reviewer #1: I am sorry but I cannot address the publication of the manuscript yet.

In the previous revisions I suggested the authors to perform changes on the factor model of the FTD-SS and to consequently adapt the results of the other analysis. The authors deleted 9 nine items from the questionnaire, which improved the fit of the model, however still kept those nine item to compute total score. How can this be correct? The readers could ask why those items were used if they were not included in the model? What is the portion of the variance explained by those items? Why did the authors not want to remove those items from the total score? The decision to still keep the nine items cannot just be theoretical. If the analyses suggest that these items should be removed, then it is pointless to compute total score with those items because they do not contribute to the factor model at all. Total score needs to be computed from only the items included in the model.

If the authors wish to compute two different total scores (one with 20 items and the other with 29 items) and analyze whether there are any statistical differences, then it is ok, but they cannot use all 29 items when the model suggested the inclusion of only 20 items.

We agree that it is better to have a more consistent approach throughout the manuscript. In fact, we did previously analyse total scores calculated both from the revised 20-item scale and the original 29-item scale, and the results of the bivariate correlation analyses between scores from the 20-item version of the scale and the other schizotypy measures were presented in S1 Supporting Information. Our failure to change the multiple regression analyses and ANOVAs in the main manuscript was merely a matter of convenience, since repeating the analyses with total scores computed from the revised 20-item scale had negligible impacts on the results. 

In particular, the effect sizes of all zero-order correlations, partial correlations, standardized regression coefficients, and R2 values for the 20-item FTD-SS-R total scores in both of the regression models (to investigate convergence with the SPQ and O-LIFE schizotypy scales) were all within 0.03 of those for the 29-item FTD-SS total scores. Indeed, we found that the correlation between total scores calculated from the 20-items and the 29-items were very large, and the changes to the analyses with demographic variables were also largely inconsequential. 

As you had suggested previously, we now refer to the 20-item scale as the revised FTD-SS (FTD-SS-R). This change is evident in the title (p. 1) and abstract (lines 20-21, p. 2), and in the results (lines 355-356, p. 19; Table 3, p. 21; lines 384-428, p. 23-24; Table 4, p. 25; Table 5, p. 26) and discussion (e.g. lines 436 – 445, p. 27). Note that, where we refer to both the FTD-SS and the FTD-SS-R, such as when making comparisons in the discussion, we include the number of items (e.g. “20-item FTD-SS-R”) for additional clarity.

We have also repeated all of the analyses conducted after the EFA and CFA so that they are performed with total scores from the 20-item FTD-SS-R. Specifically, all effect sizes, p values and confidence intervals have been re-calculated for both the multiple regression analyses (see Table 4, p. 24) and ANOVAs (see Table 5, p. 26). However, because we discuss the demographic similarities between the samples of the current study and Barrera et al.’s (2015) sample, and the influence of demographics on the FTD-SS scores in these studies, we have moved the original table of ANOVAs based on the 29-item scale to S1 Supporting Information (Table I) rather than remove it entirely. Finally, the very minor differences between the results of analyses conducted on total scores from the original 29-item scale have also been described (lines 403-414, p. 23; lines 427-428, p. 24).

---

## [Decision Letter · Decision Letter 5]

25 Nov 2022

Assessing the dimensionality of scores derived from the Revised Formal Thought Disorder Self-Report Scale in schizotypy

PONE-D-21-24412R5

Dear Dr. Sumner,

We’re pleased to inform you that your manuscript has been judged scientifically suitable for publication and will be formally accepted for publication once it meets all outstanding technical requirements.

Kind regards,

Marco Innamorati

Academic Editor

PLOS ONE

Additional Editor Comments (optional):

Reviewers' comments:

Reviewer's Responses to Questions

**Comments to the Author**

1. If the authors have adequately addressed your comments raised in a previous round of review and you feel that this manuscript is now acceptable for publication, you may indicate that here to bypass the “Comments to the Author” section, enter your conflict of interest statement in the “Confidential to Editor” section, and submit your "Accept" recommendation.

Reviewer #1: All comments have been addressed

2. Is the manuscript technically sound, and do the data support the conclusions?

Reviewer #1: Yes

3. Has the statistical analysis been performed appropriately and rigorously? 

Reviewer #1: Yes

4. Have the authors made all data underlying the findings in their manuscript fully available?

Reviewer #1: Yes

5. Is the manuscript presented in an intelligible fashion and written in standard English?

Reviewer #1: Yes

6. Review Comments to the Author

Reviewer #1: I would like to thank the authors for addressing all my comments.

I have no further comment.

7. PLOS authors have the option to publish the peer review history of their article (what does this mean?). If published, this will include your full peer review and any attached files.

Reviewer #1: No

---

## [Editor Report · Acceptance letter]

1 Dec 2022

PONE-D-21-24412R5 

Assessing the dimensionality of scores derived from the Revised Formal Thought Disorder Self-Report Scale in schizotypy 

Dear Dr. Sumner:

I'm pleased to inform you that your manuscript has been deemed suitable for publication in PLOS ONE. Congratulations! Your manuscript is now with our production department. 

Kind regards, 

on behalf of

Dr. Marco Innamorati 

Academic Editor

PLOS ONE